# Measurement Report: Black carbon properties and concentrations in Southern Sweden urban and rural air – The importance of long-range transport

Erik Ahlberg[1][*], Stina Ausmeel[1,a], Lovisa Nilsson[1], Mårten Spanne[2], Julija Pauraite[3], Jacob Klenø Nøjgaard[4,b], Michele Bertò[5], Henrik Skov[4], Pontus Roldin[1], Adam Kristensson[1], Erik Swietlicki[1], Axel Eriksson[6][*]

[1]Division of Nuclear Physics, Lund University, Box 118, 221 00 Lund, Sweden
[2]Environment Department, City of Malmö, 208 50 Malmö, Sweden
[3]Department of Environmental Research, Center for Physical Sciences and Technology, Savanorių ave. 231, 02300 Vilnius, Lithuania
[4]Department of Environmental Science, iClimate, Aarhus University, Roskilde, Denmark
[5]Laboratory of Atmospheric Chemistry, Paul Scherrer Institute (PSI), 5232 Villigen PSI, Switzerland
[6]Ergonomics and Aerosol Technology, Lund University, Box 118, 221 00 Lund, Sweden
[a]Now at: Swedish Environmental Protection Agency, 10648 Stockholm, Sweden
[b]Now at: National Research Centre for the Working Environment, 2100 Copenhagen, Denmark

*Correspondence to*: Erik Ahlberg (erik.ahlberg@nuclear.lu.se) and Axel Eriksson (axel.eriksson@design.lth.se)

**Abstract.** Soot, or black carbon (BC), aerosol is a major climate forcer with severe health effects. The impacts depend strongly on particle number concentration, size and mixing state. This work reports on two field campaigns at nearby urban and rural sites, 65 km apart, in southern Sweden during late summer 2018. BC was measured using a single particle soot photometer (SP2) and Aethalometers (AE33). Differences in BC concentrations between the sites are driven primarily by local traffic emissions. Equivalent and refractory BC mass concentrations at the urban site were on average a factor 2.2 and 2.5, with peaks during rush hour up to a factor ~4, higher than the rural background levels. The number fraction of particles containing a soot core was significantly higher in the city. BC particles at the urban site were on average smaller by mass and had less coating owing to fresh traffic emissions. The organic components of the fresh traffic plumes were similar in mass spectral signature to "hydrocarbon-like organic aerosol" (HOA), commonly associated with traffic. Despite the intense local traffic (~30 000 vehicles passing per day), PM1, including organic aerosol, was dominated by aged continental air masses even at the curbside site. The fraction of thickly coated particles at the urban site was highly correlated with the mass concentrations of all measured chemical species of PM1, consistent with aged, internally mixed aerosol. Trajectory analysis for the whole year showed that air masses arriving at the rural site from eastern Europe contained approximately double the amount of BC compared to air masses from western Europe. Furthermore, BC from the largest region emissions in the Malmö/Copenhagen urban area transported to the rural site is discernible above background levels only when precipitation events are excluded. We show that continental Europe, and not the Malmö/Copenhagen region is the major contributor to the background BC mass concentrations in southern Sweden.

# 1 Introduction

Virtually everywhere in the world, a fraction of the ambient aerosol consists of soot. Soot is formed by incomplete combustion of carbonaceous fuels at hot air-starved conditions, and commonly contains both highly absorbing graphitic-like black carbon (BC) and organic carbon. It has severe effects on climate and human health (e.g. Bond et al., 2013; WHO, 2021; IARC, 2014; IPCC, 2021). Therefore, ambient measurements of soot concentrations and properties are of great importance to constrain and model its effects. Soot measurements are complicated by the fact that the ambient aerosol is a dynamic mixture, and that soot

from different sources may not possess the same properties in terms of chemical content and nanostructure (Vander Wal et al., 2010; Malmborg et al., 2019), light absorption (Sandradewi et al. 2008), size (Schwarz et al., 2008), and toxicity (Hakkarainen et al., 2022).

A soot particle survive in the atmosphere approximately 5-9 days (Textor et al., 2006). During its lifetime several processes affects its properties. Soot is formed in a chain of steps, starting with the inception of the first condensed-phase particles from

gas-phase hydrocarbon soot precursor species (Michelsen et al 2020). These then grow rapidly by coagulation and gas-to-particle conversion and become more graphitic, also making them more light-absorbing. These near-spherical 10-30 nm primary particles coagulate to form agglomerates that are emitted from the source to the surrounding air, if not removed by oxidation already in the combustion process or by exhaust after-treatment (e.g. particle filters). The properties of the emitted soot particles depend on the combustion conditions and fuel used. For example, it is well known that the wavelength

dependence of light absorption is stronger for biomass burning BC than for traffic emissions (Sandradewi et al., 2008).

Once in the atmosphere, the freshly formed soot particles evolve from their initial agglomerated state, each consisting of large numbers of primary soot particles, into compacted soot cores with significant coatings of inorganic and organic material (Corbin et al., 2023). During such atmospheric ageing, the soot particles will not only increase their size and effective density, but also their ability to absorb light will typically increase (Zhang et al., 2018). Further, the particles change from being nearly

hydrophobic into hygroscopic particles that can act as CCN (cloud condensation nuclei; Swietlicki et al., 2008) which increases the likelihood for wet removal which is the main deposition process for BC in the atmosphere (Textor et al., 2006). This transformation depends on atmospheric conditions and constituents but can be on the order of hours under favourable conditions (Eriksson et al., 2017).

Already in 2012, the International Agency for Research on Cancer, which is part of the World Health Organization (WHO),

classified diesel engine exhaust as carcinogenic to. WHO further concluded that there is sufficient evidence for an association between short-term BC levels and all-cause and cardiovascular mortality, and cardiopulmonary hospital admissions (WHO, 2013).Drawing similar conclusions as the WHO, the US EPA (2019) summarizes the associations between several health effects and BC concentrations in their impact assessment (USEPA, 2019). In its latest update of the Air Quality Guidelines, WHO did not yet recommend air quality guidelines for black carbon (WHO, 2021). Instead, they made a statement of good

practice recommending systematic measurements of black and elemental carbon, and production of BC/EC emission inventories, exposure assessments and source apportionment.

A myriad of BC measurement techniques exist, and the terminology is based on the measured property (Petzold et al., 2013). From light absorption techniques, the equivalent black carbon (eBC) mass concentration is obtained from the ratio of measured light absorption coefficients to the corresponding mass absorption cross-section (MAC). More recently, techniques based on laser-induced incandescence (LII) have been developed for measuring the refractory black carbon (rBC) mass concentration. For instance, the single particle soot photometer (SP2) (Schwarz et al., 2006; Stephens et al., 2003) deploys LII to measure the single particle rBC mass. In addition, the SP2 can be used to retrieve the rBC core size, mass-size distribution and provide a an estimation of the coating thickness of rBC particles(Moteki & Kondo, 2007).

Although BC mass concentrations are routinely estimated from optical methods, BC number concentrations, size distributions and mixing state are rarely measured. In a previous study it was shown that global aerosol microphysics models underestimate the BC particle size, by a factor of ~2-3, while overestimating the number concentrations, by more than a factor of 3, compared to airborne measurements using the SP2 (Reddington et al., 2013). BC from different sources have different size distributions (Schwarz et al., 2008; Laborde et al., 2013; Saarikoski et al., 2021), that affects both transport (lifetime) and light interactions (Hinds, 2012), as well as deposited dose (Alfoldy et al., 2009; Rissler et al., 2012). Recent studies have pointed to the importance of BC mixing state, governed by emission source and atmospheric ageing, in understanding the light absorbing properties, and hence climatic impacts, of BC containing particles (e.g. Liu et al., 2017; Liu et al., 2019; Fierce et al., 2020; Yuan et al., 2020). These properties of BC, that are crucial to understand both health and climate impacts, are not measured by the BC measurement techniques commonly used by monitoring networks.

The aim of this study was to compare BC particle properties in nearby urban and rural settings, and to investigate the influence of urban emissions on the rural background air, utilizing both filter-based absorption measurements and single particle LII. We compare BC mass and number concentrations, size distributions and mixing state. Furthermore, the relation to total particle number concentrations and chemically resolved $PM_1$ are assessed. Trajectory analysis was used to assess the influence of long- and short-range transport.

## 2 Methods

### 2.1 Measurement sites

Consecutive aerosol measurement campaigns were performed at existing field sites in a rural and an urban setting in southern Sweden, during the period late July to early October 2018. The weather did not change dramatically between the two campaigns, hence seasonality is not expected to play a major role for the results. Figure S1 shows that the BC concentrations at the rural site during July-October did not change drastically between the times of the two campaigns. The urban campaign (September-October) took place at a curbside measurement site near a busy road junction surrounded by 4-7 story buildings (about 30 000 vehicles per day) in Malmö, which has about 300 000 inhabitants, situated about 30 km east of Copenhagen. Air was sampled at 3 meters above the pavement using ¼'' stainless steel tubing and a $PM_{2.5}$ cyclone. Total inlet length was 4-5 meters and the flow was 7-8 liters per minute. The validity of the curbside site measurements as an indicator of the BC

concentrations in the city was assessed by comparing simultaneously collected data at a rooftop urban background site (20 m a.g., PM$_{2.5}$ inlet). The rural measurements (July-August) were performed at the Hyltemossa Research Station, about 65 km northeast of Malmö/Copenhagen. This station is part of both the ACTRIS (Aerosol, Clouds, and Trace gases European Research Infrastructure) and ICOS (Integrated Carbon Observation System) European environmental research infrastructure networks. The field site is surrounded by a managed spruce forest. Aerosols were sampled at a height of 30 meters above ground through a PM$_{10}$ inlet and 25 mm stainless steel tube, at a flow rate of 16.7 liters per minute. The single particle soot measurements at the rural site were conducted through a ¼'' tube at 20 liters per minute (turbulent flow), due to simultaneous Eddy Covariance flux measurements (not described here).

## 2.2 Instrumentation and Analysis

### 2.2.1 Overview

Instrumentation to measure aerosols and gases was similar at both sites. Figure 1 shows during which time the urban and rural campaigns took place and which data was used. BC properties and concentrations was determined using a Single Particle Soot Photometer (SP2) that was moved between the sites and Aethalometers (AE33) deployed permanently. Aethalometer data for the whole year was used from the rural site to assess source regions. The particle size distribution was measured using Scanning (urban) and Differential (rural) Mobility Particle Sizers. During the latter part of the urban campaign, chemically resolved particle constituents were measured simultaneously at both sites with Soot Particle Aerosol Mass Spectrometers (SP-AMS). An Aerosol Particle Mass Analyzer (APM) was at a later stage deployed at the urban site together with a Differential Mobility Analyzer (DMA) to measure the size resolved effective density and particle mass.

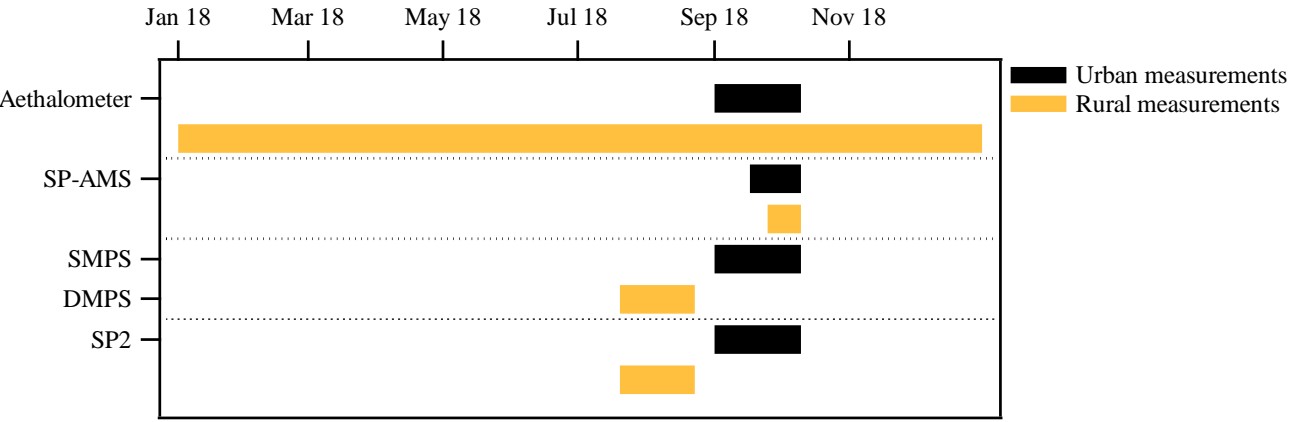

**Figure 1. Overview of the data and instruments used. Not shown in the figure are DMA-APM measurements of effective density during spring 2019.**

## 2.2.2 SP2

The SP2 (Droplet Measurement Technologies) measures the refractory black carbon (rBC) mass of individual particles using laser-induced incandescence (LII) (Stephens et al., 2003; Schwarz et al., 2006). In the SP2, particles are passed through a continuous 1064 nm Nd:YAG laser beam. Light absorbing particles (such as soot) are heated by the laser to temperatures where they will start to incandesce, if they are large enough. The incandescent light, measured by optical detectors, is linearly proportional to the BC core mass and independent of particle coating and fractal structure (Schwarz et al., 2006; Slowik et al., 2007). The approximate range of the SP2 is 0.5 – 200 fg per particle, where the lower limit is due to energy dissipation, and the particles in upper limit may either saturate the detectors or not absorb enough energy to incandesce. The overall relative uncertainty in mass concentration by the SP2 was estimated by Sharma et al. (2017) to be within 25-38 %. To limit the data collected, not all particle peak signals were saved (10 and 1-5 % was saved in the rural and urban campaigns respectively).

The SP2 incandescence channels were calibrated using monodisperse Aquadag (provided by Droplet Measurement Technologies) before, during and after the field campaigns. Particle sizes were selected using a DMA and mass was calculated using the effective density recommended by Gysel et al. (2011). Since the calibration during the rural campaign did not yield satisfactory data, the calibration during the urban campaign was used to analyze both datasets. Although this is not optimal, the incandescence detector calibration is relatively stable in time. The response of the broadband incandescence detector from typical ambient BC particle masses varied with less than ±15 % for calibrations performed before and after the rural campaign. The SP2 sensitivity to ambient BC has been shown to resemble that of fullerene soot particles (Baumgardner et al., 2012). Following Laborde et al. (2012b), the Aquadag calibration factors were therefore translated into a fullerene soot equivalent calibration using a scaling factor of 1.34 at 8.9 fg and an intercept of 0.

The calibrations and campaign data were processed using the PSI SP2 toolkit version 4.111. BC particles between mass equivalent diameters of 64-580 nm (using a density of 1800 kg m-3) were included in the result. The lower size was selected from looking at the sharp decrease in the number-size distribution when including very low peak heights. The mass and number concentrations presented are not corrected for particles outside this range. Pileci et al. (2021) found that the rBC mass outside the range of the SP2 is on the order of 20 % for several European background stations, depending on the size distribution. No particle detection efficiency calibration was performed. Lognormal fits to averages of 2 and 24h were used to calculate geometric mean diameters.

BC particle coatings can be estimated by the SP2 using a separate scattering detector. The delay-time method was used to estimate BC particle coatings in a qualitative way (Moteki and Kondo, 2007; Subramanian et al., 2010). This method separates BC particles as being either thinly or thickly coated depending on the delay-time between the maximum scattering and incandescence peak heights. Coated BC particles generate two scattering peaks, one from the evaporating coating and one from the BC core itself. The scattering signal of the coating will be detected before the incandescence signal since BC particles take some time to heat up to the point of incandescence, and absorbed energy will dissipate through evaporation of the coating material. For thickly coated particles, the scattering of the coating will exceed the scattering of the BC core, while for thinly

coated particles the opposite is true. Moteki and Kondo (2007) and Laborde et al. (2012a) showed that a coating volume fraction of 70 % is needed for a particle to be classified as thickly coated by the SP2.

### 2.2.3 Aethalometer

Aerosol light absorption was measured at both sites using a seven wavelength Aethalometer (model AE33, Magee Scientific) (Drinovec et al., 2015). Equivalent black carbon (eBC) mass concentrations were calculated using absorption at 880 nm and the default mass absorption cross section (MAC) value of 7.77 $m^2$ $g^{-1}$. The absorption Ångström exponent (AAE), a measure of the spectral dependence of light absorption and commonly used in source apportionment of BC (Kirchstetter et al., 2004; Sandradewi et al., 2008), was calculated between the wavelengths 370 - 950 nm. The relative uncertainty of the absorption coefficient measured by the aethalometer increases with lower BC mass concentrations, and has been shown to converge around 30 % for an older version of the instrument (Backman et al., 2017). The uncertainty of eBC also includes the uncertainty in the MAC value which was not estimated in the present study.

### 2.2.4 SP-AMS

The chemical composition of refractory and non-refractory particulate mass was measured in real-time with two soot particle aerosol mass spectrometers (SP-AMS, Aerodyne Research Inc.) (Onasch et al., 2012), at the urban and rural measurement site, respectively. The measurements were performed in parallel, and there are 15 days of overlap (Sept. 25 to Oct. 10). The SP-AMS detects particles within the vacuum aerodynamic diameter size range of about 70 nm to 1 µm. The aerosol sample flow is focused through an aerodynamic lens. The (non-refractory) components of particles in the air beam are vaporized on a heated tungsten plate (600 ℃), subjected to 70 eV electron ionization, and the ions produced are detected and categorized (organic, nitrate, sulfate, ammonium or chloride) by a high-resolution time-of-flight mass spectrometer. In the SP-AMS set-up a laser is used to also vaporize the refractory BC (rBC). We operated both SP-AMSs in the "dual vaporizer" configuration as further discussed in the SI. Calibrations were performed in the field using 300 nm mobility diameter particles from nebulizing ammonium nitrate as described in Onasch et al (2012). The absolute concentration of species at the urban site are not shown due to uncertainties in the calibration and dissimilarities when comparing to other instruments. For the rural data, $PM_1$ derived from the DMPS (using a density of 1.5 g $cm^{-3}$) and a Palas Fidas 200 at a nearby site agreed well with that from the SP-AMS. All rBC data presented in this work are derived from the SP2 measurements which are more accurate. Data analysis was performed with IGOR Pro 7 (Wavemetrics, USA), and the AMS analysis software package including SQUIRREL 1.61F, and PIKA 1.21F.

For the urban dataset, the fresh traffic aerosol was isolated through selection of distinct traffic plumes, i.e. short (seconds to minutes) increase in concentration. Increases in rBC number concentration from the SP2 and the concentration of the HOA marker ion $C_4H_9^+$ from the AMS were used as indicators of fresh traffic plumes, and 90 such plumes were selected. For each plume, the current urban background aerosol was defined by selecting two windows, one before and one after the plume, of about 2 minutes, i.e. a total of 4 minutes (see Figure S2). The typical plume duration was about 30-60 seconds. Organic aerosol

(OA) mass spectra from the AMS were calculated for the plume and for the background respectively, and the background spectra were subtracted from the plume spectra to obtain a net contribution to OA mass concentrations. The m/z tracer method (Ng et al., 2011) was used to estimate the different OA components.

### 2.2.5 DMPS/SMPS

Particle size distributions were measured between electrical mobility diameters of 3.2 - 900 nm at the rural site and 11.5 – 604
nm at the urban site. At the rural site the DMPS consisted of of two Hauke type DMAs and two condensation particle counters (CPC, TSI 3772 and TSI UCPC 3025) (Wiedensohler et al., 2012). At the urban site the SMPS consisted of a TSI DMA 3082 and a TSI CPC 3772. At both sites, the aerosol was dried before sizing using Permapure driers with low pressure sheath air. Size distributions at both sites were averaged over 1 hour.

### 2.2.6 DMA-APM

Size dependent particle mass and effective density were measured with a DMA-APM. Pre-selection of particle size was done with a DMA (TSI 3082), which selects particles based on electrical mobility diameter. The quasi-monodisperse aerosol was then led through the APM (model 3600, Kanomax), which measures the relationship between electrical mobility and particle mass. In the APM, the aerosol passes a rotating cylinder with an applied voltage, and the particle mass is determined based on the balance equation between the centrifugal force and electrostatic force (McMurry et al., 2002). The mass selected particles
were counted after the DMA-APM with a CPC (TSI 3772). The aerosol flow was 1 liter per minute. The effective density is calculated based on the electrical mobility diameter and the particle mass from the DMA-APM. For spherical particles, this is the bulk density, but for non-spherical particles, the effective density depends on the particle shape factor. Six different particle mobility diameters were tested, 50, 75, 100, 150, 250, and 350 nm. The APM was operated in a constant RPM mode, where the angular rotation speed was constant and the APM voltage was increased step-wise during a time period of 15 minutes per
scan. The rotational speed and voltage were selected so that the effective density was measured in approximately the range 0.1-3.5 g cm$^{-3}$. The APM was run during five days in spring 2019, after the intensive campaigns to verify the bimodality, in terms of effective density, of the city aerosol.

### 2.2.7 Particle inlet losses

Particle losses in the inlets were assessed using the Particle loss calculator (von der Weiden et al., 2009). The losses in the
longest tube sections of each inlet that was used are shown in Fig. S3. Losses in additional tubing and the actual inlets are expected to be similar (and small) in all cases. In the laminar flow inlets at both sites, PM$_1$ losses are negligible, while the turbulent flow inlet has losses of 20-30 % for spherical particles with a diameter between 100 – 1000 nm (density 1600 kg m$^{-3}$). Based on these calculations and the measured rBC size distribution (discussed below), the SP2 mass and number concentrations measured at the rural site have been adjusted to correct for 25 and 30 % losses respectively since adjusting by
size is not possible because the size of rBC cores including coating was not measured.

### 2.2.8 Trajectory analysis

Wind direction and speed data from the rural site at Hyltemossa were downloaded from the ICOS Carbon portal (ICOS 2019), and analyzed using the Igor pro tool Zefir (Petit et al., 2017). To investigate how the air mass origin influence the BC concentrations at the rural site, we used the Hybrid Single Particle Lagrangian Integrated Trajectory Model (HYSPLIT) with
meteorological data from the Global Data Assimilation System (GDAS) (Stein et al., 2015; Rolph et al., 2017). Seven days HYSPLIT backward trajectories, arriving with one hour intervals at 100 m a.g.l. at Hyltemossa, were calculated for year 2018. The air masses were then classified into four different categories depending on which geographical region the air masses spent most time over. For this classification we only considered the air mass origin during the last 48 hours upwind Hyltemossa and the cumulative time which the transported air parcels were below 1000 m a.g.l.. Figure S4 in the supplementary material
illustrate how the air mass origin were divided into North Westerly (NW), South Westerly (SW), South Easterly (SE) and North Easterly (NE). In addition we separated the air mass origin depending on if they passed over the Malmö/Copenhagen region less than 12 hours before the air masses arrived at Hyltemossa.

The long-distance transported BC source contributions during the rural and urban measurement campaigns were estimated using the Lagrangian 1D-column chemistry transport model ADCHEM (Roldin et al., 2011; Roldin et al., 2019). ADCHEM
was setup and run forward in time along pre-computed 14-days backward HYSPLIT air mass trajectories arriving 100 m a.g.l. at Hyltemossa, one new trajectory every hour. Source specific anthropogenic BC emissions along the trajectories were taken from the CAMS-GLOB-ANT v4.2 global emission inventory, which has a spatial resolution of 0.1°x0.1° (Granier et al., 2019). BC emissions from wildfires were estimated using the GFED4 emission inventory (Randerson et al., 2018).

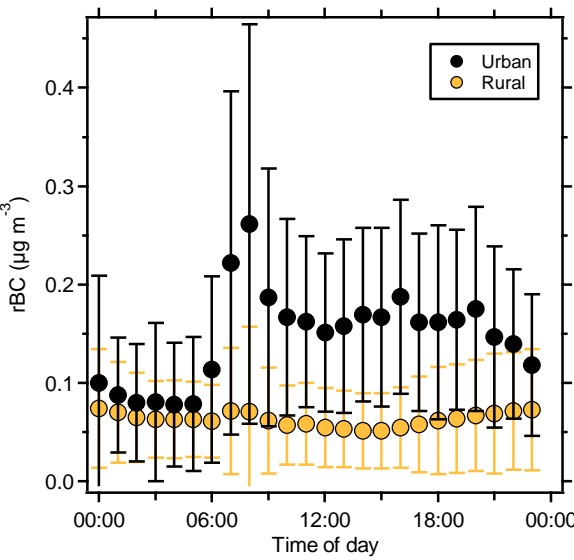

**Figure 2. Diurnal cycles of rBC mass concentrations (±1σ) from the urban and rural sites as measured by the SP2. Values are averages of data collected during 34 (rural) and 41 (urban) days of sampling.**

# 3 Results and Discussion

## 3.1 BC concentrations

An overview of the BC measurement results is shown in Table 1. As expected, the curbside site has higher concentrations than
the rural site, with campaign average factors of 2.5 for rBC mass, 2.2 for eBC mass, and 3.2 for rBC number concentration.
The differences in concentrations are due to local emissions, clearly shown in the diurnal pattern (Fig. 2), which follows traffic
intensity. The comparison between the urban street-level and urban background eBC levels are very similar in time-series but
with a lower daily maximum for the roof-top measurements (Fig. S5). This suggests that the curbside measurements are
indicative for the city. The average eBC mass concentration at the roof-top site during the campaign was 15% lower than at
the curbside site. The largest difference between the urban and rural sites occur during morning rush hour when rBC mass
concentrations are up to a factor 3.7 higher in the city. The mass concentrations are still higher in the city during nighttime,
before the first rush hour peaks, but only by a factor of 1.2-1.3. The weekends have lower rBC mass concentration ($M_{rBC}$ of
$0.11 \pm 0.06$ µg m$^{-3}$) compared to weekdays in the city, but is still a factor of 1.8 higher than the average for the rural background
station. The AAE is slightly higher at the urban site, which could be due to the differences in BC size distribution and coating
(Virkkula, 2021).

**Table 1. Averages ±1σ (original averaging time in parenthesis) for the urban and rural campaigns. rBC mass (M) and number (N) concentrations are from SP2 measurements and eBC mass concentrations are from absorption at 880 nm using a MAC of 7.77 m$^2$ g$^{-1}$. Mass-based geometric mean diameter (GMD$_M$) and geometric standard deviation (GSD$_M$) are from the rBC mass equivalent size distribution. rBC concentrations are not corrected for particles outside the range of the instrument. Rural rBC concentrations**
**are corrected for inlet losses. Absorption Ångström exponents (AAE) are calculated between absorption at 370 and 950 nm.**

|  | Rural | Urban |
| --- | --- | --- |
| $M_{rBC}$ (1h), µg m$^{-3}$ | 0.06 ±0.05 | 0.15 ±0.11 |
| $M_{eBC}$ (1h), µg m$^{-3}$ | 0.22 ±0.16 | 0.49 ±0.45 |
| $N_{rBC}$ (1h), cm$^{-3}$ | 31 ±21 | 100 ±80 |
| GMD$_M$ (24h), nm | 168 ±12 | 141 ±11 |
| GSD$_M$ (24h), nm | 1.74±0.08 | 1.77±0.04 |
| AAE$_{370-950\ nm}$ (1h) | 1.13 ±0.12 | 1.24 ±0.26 |

## 3.2 Estimation of BC particles number fractions

The number fraction of particles containing a BC core was estimated by comparing the daily average of BC number
concentration from the SP2 to the number concentration of particles larger than 64 nm as measured by the mobility particle
sizers. At the rural site this fraction was $2.7 \pm 0.6$ (1σ) %, while for the urban site the fraction was $13.4 \pm 5.5$ (1σ) %. These
estimates are biased low because the BC size distribution (as measured by a mobility particle sizer) is shifted to larger sizes if
shape factors and coating is added. Increasing the cut off size for total particle number measured by the SMPS to $D_P > 130$ nm
(assuming a coating thickness of similar magnitude as the rBC core mass equivalent diameter) the fractions are increased to 6
% in the rural campaign and 45 % in the urban campaign. In a study outside Paris (Laborde et al., 2013), the number of BC
particles counted by the SP2 (with no correction for particles outside the measurement range) was 0-15 % of the total particle

number concentration (above 20 nm) measured by an SMPS. In airborne measurements over Europe (using an SP2 and a passive cavity aerosol spectrometer probe), Reddington et al. (2013) found that the number fraction of approximately 14 % of particles with dry diameters above 260 nm contained a BC core. In an older study, Rose et al. (2006), using a volatility tandem-DMA set-up, measured summertime BC particle number fractions of 2-7 % at a rural site and 20-60 % in a street canyon, depending on the selected size (30-150 nm in mobility diameter).

### 3.3 rBC vs. eBC

The difference between rBC and eBC is significant at both sites (Fig. S6). For the full campaign averages, eBC is more than 3 times higher than rBC at both sites, with median ratios (eBC/rBC) of 3.04 and 3.65 the urban and rural sites respectively. Several studies have found similar discrepancies between rBC and eBC (Holder et al., 2014; Raatikainen et al., 2015; Sharma et al., 2017; Tasoglou et al., 2018; Li et al., 2019). The rBC mass in this study is biased low because of the limited range of the SP2, but given the size distributions measured (example shown in Fig. S7), this mass bias should be on the order of 10-20 %, which still only would lower the ratio between eBC and rBC to 2.4-2.7. One reason for biased high eBC values is that the measured absorption can be amplified by non-absorbing coatings of BC cores (Kalbermatter et al., 2022). It is well known that site specific MAC can differ from the default values from the manufacturer of the AE33 (e.g. Cui et al., 2016). At a site nearby the rural site of this study, a MAC of 12.64 $m^2$ $g^{-1}$ at 880 nm has previously been derived using aethalometer (AE33) absorption coefficients and elemental carbon (EC) measurements (Martinsson et al., 2017). This value is 1.6 times higher than the default and if used, the ratio between eBC and rBC of the rural campaign would be ~2.

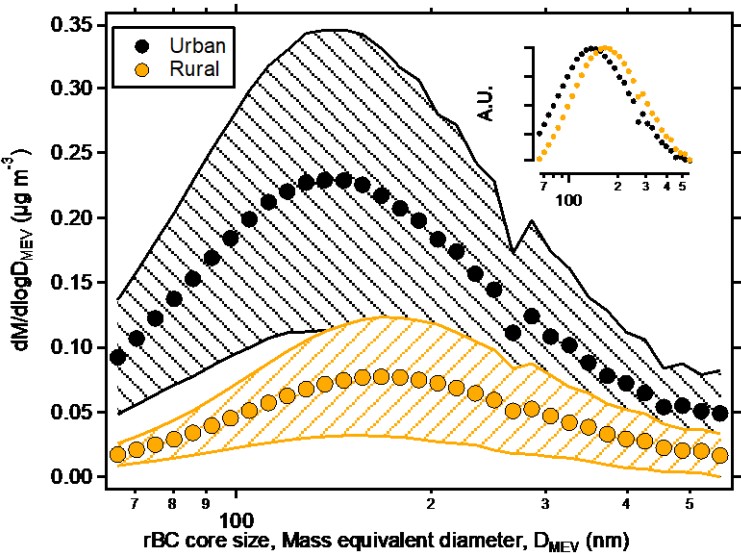

**Figure 3. Average mass-size distribution of BC cores for the urban (black) and rural (orange) campaigns. Filled areas show ±1σ. Insert shows the normalized distribution for comparison of the modes. Dip at 270 nm is due to erroneous stitching of high and low gain detector channels.**

### 3.4 BC core size and coating

The average geometric mean diameter (GMD) of the rBC mass size distribution, from lognormal fits of 24 h data, was highest in the rural setting (Table 1). The average mass size distributions for the full campaigns are shown in Fig. 3. Mass equivalent

diameters of 168 and 148 nm for the rural and urban sites respectively, correspond to BC particle masses of 4.5 and 3.1 fg. The number size distribution generally peaked close to the SP2 detection limit (64 nm, corresponding to 0.25 fg, for both campaigns), and the GMD of this distribution is therefore more uncertain, and not presented here. An example of 24 h number and mass size distributions are shown in Fig. S7. The difference in mass GMD is statistically significant (P<0.01) and can be explained by different BC particle sources, air masses and coagulation. Figure S1 shows that while BC concentrations at the

rural site were similar during the two campaigns, the source sectors changed slightly. Notably, more industrial emissions and less shipping emissions affected the concentrations during the urban campaign. Figure 4 shows the diurnal variation in GMD from mass size distributions averaged over 2 h. The urban site has a higher fraction of particles coming from local traffic, which have been shown to be smaller than particles from e.g. biomass burning (Schwarz et al., 2008; Laborde et al., 2013; Saarikoski et al., 2021). This shifts the size distribution during daytime to slightly lower sizes, although the measurement

uncertainties, including calibrations, of the SP2 are close to the difference in average GMD between the sites.

The BC particles at the rural site were characterized by a thicker coating than at the urban location, as measured by the SP2 delay time method (a typical example is shown in Fig. S8). The fraction of "thickly coated" BC particles with an incandescence peak height corresponding to a BC core diameter of mass 2.4 fg (~136 nm mass equivalent diameter) was on average $71 \pm 8$ % ($1\sigma$) at the rural site and $38 \pm 7$ % ($1\sigma$) at the urban site, with no big difference during weekends. The short-term APM

measurements verified the bimodal structure in effective density of the urban aerosol. The mass distribution of 150 nm particles is shown in Fig. 5. The two modes suggest that the aerosol is externally mixed, with more and less dense particles present. The two lognormal curve fittings have a mean mass of $0.86 \pm 0.02$ fg, and $2.49 \pm 0.02$ fg, respectively. These masses correspond to effective densities of $0.49 \pm 0.01$ g cm$^{-3}$, and $1.41 \pm 0.01$ g cm$^{-3}$, respectively, which is very similar to the effective densities presented by Rissler et al. (2014), who reported two effective densities, for 150 nm particles sampled in downtown

Copenhagen, of $0.53 \pm 0.05$ g cm$^{-3}$, and $1.36 \pm 0.15$ g cm$^{-3}$, respectively. For 50 and 75 nm particles, the average mass distribution was unimodal with a mean mass of $0.062 \pm 0.001$ fg and $0.26 \pm 0.01$ fg, respectively, corresponding to effective densities of $0.95 \pm 0.01$ g cm-3 and $1.19 \pm 0.02$ g cm-3. The average mass distribution of 100 nm particles was bimodal with mean masses of $0.33 \pm 0.01$ fg and $0.68 \pm 0.01$ fg, corresponding to effective densities of $0.63 \pm 0.02$ g cm-3 and $1.29 \pm 0.01$ g cm-3. These values are also similar to the densities in Rissler et al. (2014), but we did not observe an additional mode at

higher masses for 50 and 75 nm particles, but only the less dense, fresh soot mode. However, in single APM spectra, the mean mass mode was sometimes different from the average, and on a couple of occasions, a bimodal distribution was observed also for the smaller particles. No mass mode could be extracted for APM measurements of larger particle sizes (250 and 350 nm) due to low particle concentrations and consequently poor counting statistics.

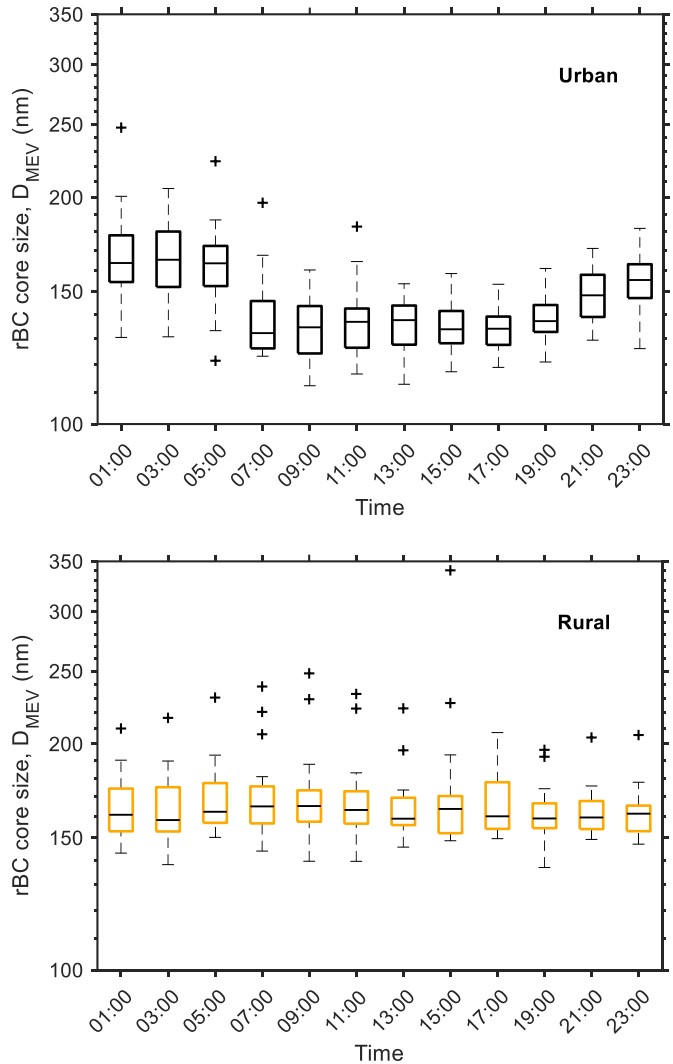


**Figure 4. Box-plot showing the daily pattern of rBC GMD (2 h averages) as measured by the SP2, at the urban and rural sites. Boxes show the median, 25th and 75th percentiles, with values more than 1.5 below or above those considered outliers (+).**

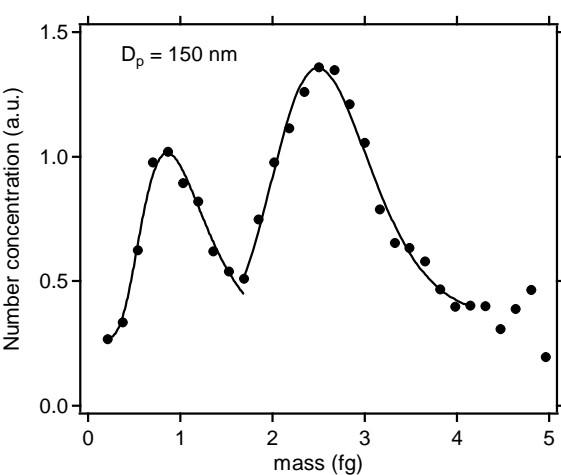

**Figure 5. Measured DMA-APM spectrum for urban particles with a diameter of 150 nm (black markers). The spectrum is the average number concentration (arbitrary units) of eleven samples from all sampling days. Two lognormal fits are shown in black.**

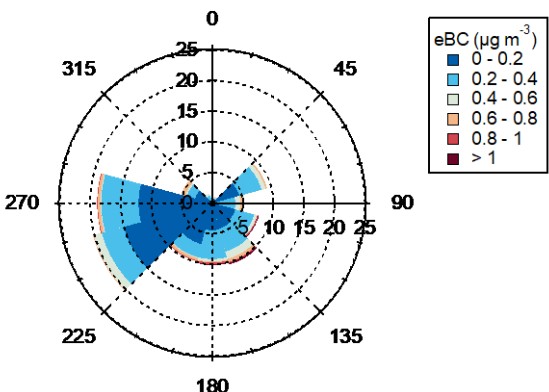

**Figure 6. Wind rose showing the frequency of different wind directions (30° resolution) during the rural campaign and corresponding concentrations of eBC.**

### 3.6 Origins of BC in the rural background air

Figure 6 shows the wind direction probability together with eBC from the five weeks rural campaign. The Malmö/Copenhagen area is in the direction 200 – 230 degrees as seen from the rural site. Based on the HYSPLIT trajectory analysis (Sect. 2.2.8) , some peaks in eBC, non-refractory PM1, and total particle number concentrations at the rural site coincide with air masses originating over Malmö/Copenhagen. These urban plumes were discernible when the air mass was relatively clean (typically of western or north-western origin). However, no conclusive results on the influence of the urban site on rural levels could be derived from analysis of the limited field campaign data. Also when analyzing the complete eBC dataset from 2018 there is no significant difference in the median eBC concentration for air masses with and without influence from Copenhagen and/or Malmö, according to the trajectory analysis. However, if we only consider air masses with insignificant precipitation within

48 hours upwind the rural site (i.e. less than <1 mm precipitation according the HYSPLIT model), the measured eBC median concentration is significantly higher in the air masses that have moved over Copenhagen/Malmö, 354.7 ng m-3 compared to

226.9 ng m-3 without influence from Copenhagen/Malmö. Similar results are also found when only the air masses originating from SW is considered (see Fig. 7 and Tables S2-S4). Most likely the contribution from the BC emissions from Copenhagen/Malmö is not apparent when analyzing the whole eBC dataset from 2018 because the SW air masses are generally influenced by more precipitation (more BC wet scavenging) compared to other air masses. The HYSPLIT trajectory simulations for the complete year show that 66 % of all air masses that passed over Malmö/Copenhagen were influenced by

>1 mm precipitation within 48 hours upwind the rural site, compared to 39 % for all other air masses. The lowest median eBC concentration is found in air masses from NW (101 ng m-3) followed by NE air masses (149.9 ng m-3) and SW air masses (192.0 ng m-3). SE air masses clearly stand out from any other air masses with a median eBC concentration of 563 ng m-3. This can partly be explained by the low probability of precipitation in SE air masses. Only 23 % of these air masses are influenced by >1 mm precipitation within 48 hours upwind the rural site. However, also when considering the effect of

precipitation upwind Hyltemossa the SE air masses have a median eBC concentration that is ~2 times larger than the SW air masses.

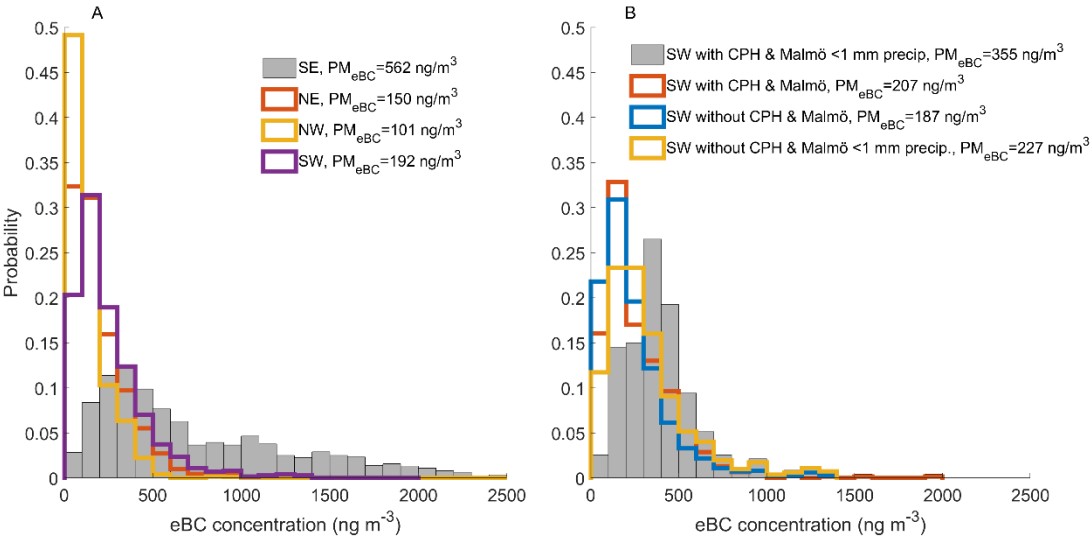

**Figure 7. Trajectory analysis of different air mass origins, with corresponding median eBC mass concentrations in the legend. Panel A shows histograms with the observed eBC from the Aethalometer at Hyltemossa from year 2018 for air masses originating from**

**SE, NE, NW and SW as defined in Fig. S4. Panel B shows the eBC histogram from all SW air masses with or without influence from the Copenhagen (CPH) and Malmö region.**

Comparison of the 15 days of simultaneous urban and rural measurements are shown in Fig. 8. It is clear that non refractory-PM$_1$ concentration and composition are driven by long-range pollution at both sites, while black carbon is more influenced by local emissions. The organic aerosol at the urban site was clearly dominated by oxygenated organic aerosol (OOA, 78%) with

only a minor contribution of HOA (9%), suggesting that aged long-range sources dominate even the organic aerosol, for which one could expect a larger difference between the sites owing to the high urban traffic density (Glasius et al., 2011). However, in the original time resolution of 20 seconds (not shown) the urban AMS data features transient plumes of traffic exhaust. Yet, as shown in Fig. 8, these plumes have little impact on average concentration. The OA mass spectrum from traffic plumes at the curbside site is shown in Fig. S9. . The plume spectra are similar in mass spectral profiles to "hydrocarbon like organic
aerosol" HOA, commonly associated with traffic emissions. This OA is likely present in thin coatings on the freshly emitted traffic BC, and possibly also externally mixed with the BC, and hence not detected by the SP2 time delay method.

With regards to eBC, Fig. 8 shows that approximately half of the mass at the urban site is due to long range transport, despite the high traffic intensity (about 30 000 vehicles per day). This can be deduced from the temporal variability and absolute concentrations at both sites. The result is corroborated by the rBC results shown in Fig. 2. Firstly, rBC concentrations are
roughly doubled at the urban site compared to the rural site. Secondly, the diurnal pattern observed at the urban site (see Figure 2) shows that levels between 01.00 and 05.00 local time, during which local traffic density is low, are about half of the average concentration.

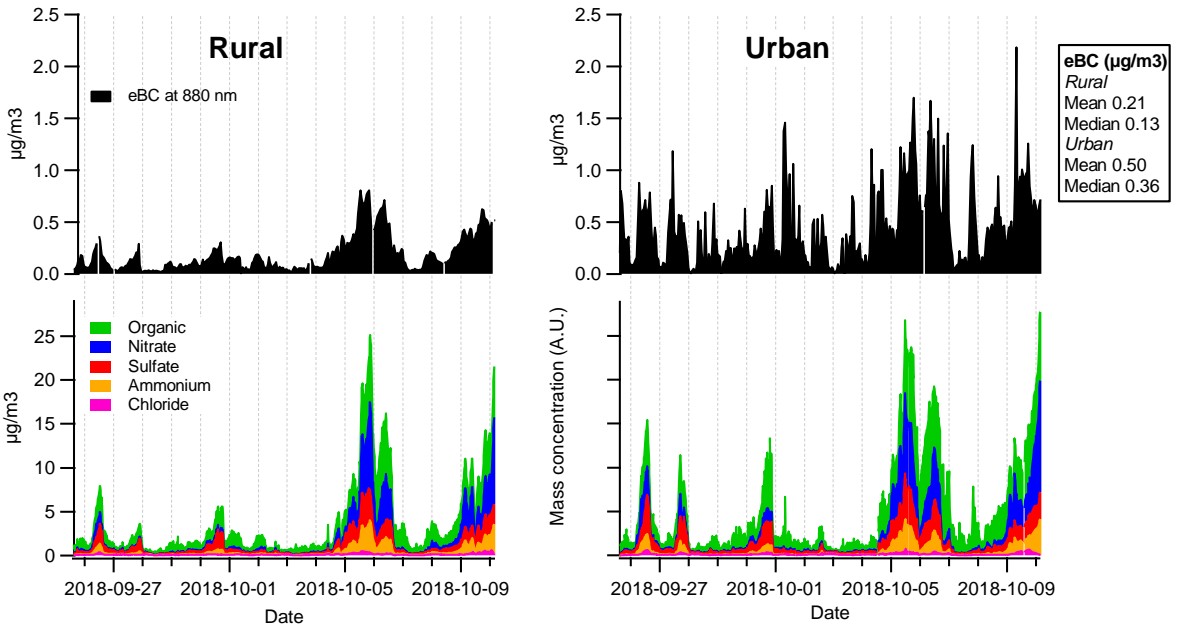

**Figure 8. Concentrations of eBC and PM$_1$ chemical species at both sites during overlapping urban and rural measurement period. 1 hour averages for eBC, 20 minutes for PM$_1$ chemical species. Absolute concentrations of PM$_1$ at the urban site was not possible to extract due to uncertainties in the calibration.**

Consistent with the high abundance of BC from long range transport at the urban site, on average 40 % of urban BC particles with a diameter close to the mass GMD were found to be thickly coated based on the SP2 data. We did not directly measure
the chemical composition of the thick coatings. However, as shown in Fig. 9, non-refractory PM$_1$ concentration is highly

correlated with thickly coated BC fraction ($R^2 = 0.87$). Indeed, all major chemical species in non refractory-$PM_1$ correlate well with the thickly coated BC fraction ($R^2 = 0.82 - 0.88$). This suggests that the BC coatings are similar in composition to non refractory-$PM_1$ (dominated by secondary material), despite the fact that only a minor fraction of the particles in $PM_1$ contain rBC as discussed above. As non refractory-$PM_1$ is more widely and accurately measured than BC coating composition such

homogeneity simplifies assessment of BC properties and impacts, for example concerning cloud formation and lung deposition. However, direct measurements of BC coating composition, ideally on a single particle basis, are needed to support this conjecture.

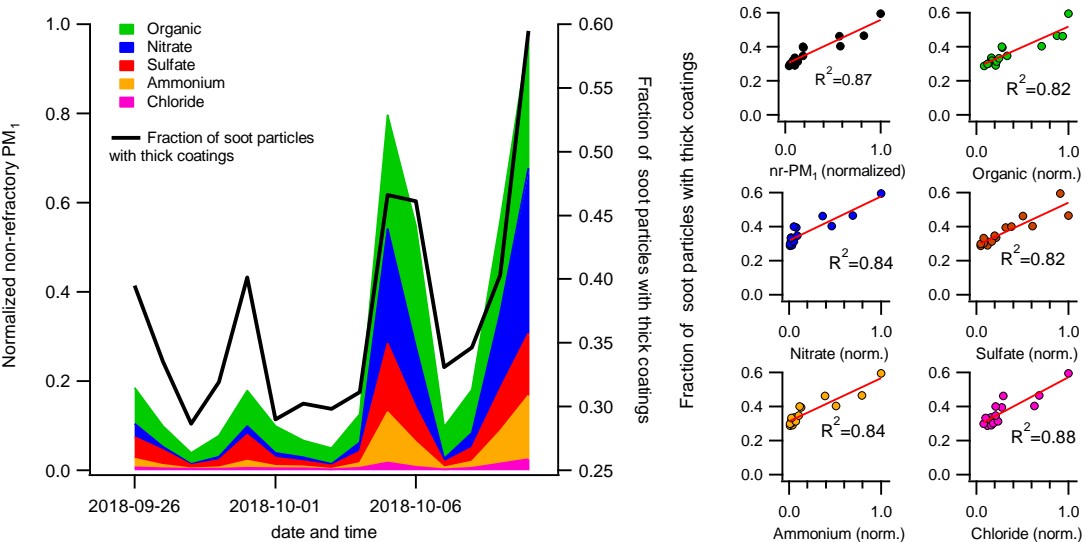

**Figure 9. Comparison of chemically resolved non-refractory $PM_1$ concentration and the fraction of thickly coated BC cores, with a mass equivalent diameter corresponding to ~136 nm, in the urban setting. Left: time series. Right: scatterplots of thickly coated BC particle fraction vs non refractory-$PM_1$ and subcategories thereof. 24 hour averages.**

## 4 Summary and Conclusions

During 11 weeks in 2018, BC particles were sampled at two nearby sites in southern Sweden. Campaign average mass

concentrations of BC, measured with two different methods, was a factor of 2.2-2.5 higher at the urban site, compared to the rural background site. Hourly averages during rush hour peaks were up to a factor 4 higher in the city. Despite good correlation between rBC and eBC, a factor of 3 higher concentrations was consistently measured with the optical method. The number fraction of particles containing BC, compared to total particle concentration above 64 nm in mobility diameter, was 2.4 and 13.4 % at the rural and urban site respectively. However, these numbers depend strongly on the sizes that are integrated and

would ideally be measured and reported in a size-segregated manner using pre-selection of particles. The BC particle size distributions were similar, with peak mode diameters slightly smaller in the city, which can be expected since a larger fraction

of the particles are from fresh traffic emissions. This was also seen in the BC particle coating, which was assessed in a qualitative manner. The rural site had approximately double the amount of thickly coated particles compared to the street measurements.

Simultaneous measurements using SP-AMS at both sites showed similarities in both time-series and mass spectra of the organic aerosol, verifying the impact of long range aerosol and secondary aerosol production. Primary organic aerosol mass spectra were investigated based on transient plumes from local traffic at the urban site, but composed a small fraction of the total organic aerosol. The fraction of thickly coated particles at the urban site was highly correlated to all measured species of the AMS, again showing the importance of long range transport.

Plumes from the nearby urban site to the rural site were not clearly distinguishable during the field campaigns. Trajectory analysis of the full year of 2018 show that significant increases in eBC concentrations at the rural site for air masses passing over the Malmö/Copenhagen area are only clearly seen during days with low precipitation. This increase, however, is small in comparison with the influence of long-range transported BC. Transport of BC from continental, and especially eastern, Europe is what governs the BC concentrations in southern Sweden background air, when looking at eBC from the full year of 2018.

**Data availability**

Processed data for main results are available at https://doi.org/10.5281/zenodo.6559236. EBC and DMPS data from the rural site are available at http://ebas.nilu.no. Raw or specific data sets are available from the authors upon request.

**Author contribution**

EA: Conceptualization, Methodology, Validation, Formal analysis, Investigation, Data curation, Writing- original draft,
Visualization, Project administration. SA: Validation, Formal analysis, Investigation, Data curation, Writing- original draft, Writing- Review & editing. LN: Formal analysis, Writing- review & editing, Visualization. MS: Investigation, Resources, Data curation, Writing- review & editing. JP: Formal analysis, Writing- review & editing, Visualization. JKN: Resources, Writing- review & editing. MB: Writing- review & editing. HS: Writing- review & editing. PR: Methodology, Software, Formal analysis, Resources, Data curation, Writing- Original draft, Writing- review & editing, Visualization, Funding
acquisition. AK: Formal analysis, Data curation, Writing- review & editing. ES: Conceptualization, Methodology, Resources, Writing- review & editing, Supervision, Project administration, Funding acquisition. AE: Conceptualization, Methodology, Validation, Formal analysis, Investigation, Data curation, Writing- Original draft, Writing- review & editing, Visualization.

**Competing interests**

The authors declare that they have no conflict of interest.

## Acknowledgements

The authors would like to thank ICOS Sweden, especially Tobias Biermann, Michal Heliasz, Jutta Holst, and Thomas Holst, for meteorological data and Hyltemossa station management, Paul Hansson at Malmö Environment Department for help with setting up the urban station measurements and Dalaplan station technical assistance. We also wish to acknowledge the support from the Swedish Research Council (Vetenskapsrådet) for ACTRIS Sweden under contract 2021-00177.

## Financial Support

This work was supported by the Swedish research councils Formas (projects 2015-00994 and 2018-01745) and VR (projects 2019-05062 and 2019- 05006).

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
