# Peer review of "Measurement Report: Black carbon properties and concentrations in Southern Sweden urban and rural air – The importance of long-range transport"

_Atmospheric Chemistry and Physics, 2022_

## Author Comment (AC1)

We sincerely thank both reviewers for thoroughly reading and commenting on our manuscript. Our answers, in normal text, are presented below the comments, which are bold. Changes to the manuscript are in italic with line numbers of the modified manuscript.

**REVIEWER #1**

**The work presented by Ahlberg and coauthors investigates the variability of black carbon properties at an urban and rural site in southern Sweden. This manuscript fulfils the requirements of a "Measurement report" since it treats specific aerosol measurements and processes confined in a restricted area and time. However, the manuscript is structured as a full "Research Article" and falls a bit short on certain aspects of data analysis and interpretation. The final consequence is that the motivations, methods and goals are not always clear. I recommend major revision and resubmission. I hope that the major and specific comments listed below will help the authors in the rebuttal process.**

Regarding the manuscript type, it is our understanding that the structure of a measurement report is very similar to a research article. If certain sections should be modified we hope to get feedback from the editor. With the answers below we hope that the methods and goals have been clarified.

**MAJOR COMMENTS**

**First, there is an evident problem with sections and subsections. The article is organized without subsections; thus the assimilation of the scientific message becomes particularly complicated. The structure should be modified including subsections in the "methods" and result "sections".**

We agree and have included subsections in section 2.2 (Instrumentation) and in the results in the modified manuscript to help the readers.

**The overall motivation behind the paper is unclear. The authors mentioned climatic and health implications, but these are described very generally without a local (Swedish) perspective. The manuscript does not provide enough measurement time to address climatic issues but could draw a nice, even if very short, picture of air quality. I think this should be the redline of the entire manuscript and should include as motivation: the health impact of aerosol emission in Sweden (e.g. death per year), the history of Swedish reduction strategies and subsequent effects.**

Although all field campaigns are merely snapshots of current air quality, we believe the reported results on BC size and number are the first of its kind in the region, and are therefore important to report for their influence on climate. Further, the importance of long-range transport for Swedish background concentrations are shown using a longer dataset. An epidemiological study on BC emissions in Sweden vs long range transport would be interesting but is a totally different study that could build on our results. To make the overall motivation behind the study clearer, we have changed the last paragraph of the introduction:

*L52. The aim of this study was to compare BC particle properties in nearby urban and rural settings, and to investigate the influence of urban emissions on the rural background air. BC was measured using single particle laser induced incandescence and filter-based absorption. We compare BC mass and number concentrations, size distributions and mixing state. Furthermore, the relation to total particle number concentrations and chemically resolved PM1 are assessed. Trajectory analysis was used to assess the influence of long- and short-range transport.*

**There is nothing that can be practically done about this, but the fact that the urban and background measurements are not simultaneous is the weakest point of the manuscript. The authors should convince the reader that the background aerosol population do not change drastically from July to October. Till that point, the results shown in Figures 1,2,3 could be affected by many atmospheric processes such as precipitation, and changes in emission large scale circulation. Potentially due to this problem, the goal of the authors is not always clear.**

We agree that it is unfortunate that all measurements could not be performed simultaneously since only one SP2 instrument was available. But to us, Figure 1 and 3 clearly shows that the diel cycle of BC properties in the urban environment is due to the local emissions. To compare the two periods further, we added the following

figure to the supplement, showing observed and modelled eBC at the rural background station during both the rural and urban campaign. Sources were modelled using ADCHEM (described in the manuscript) and standard CAMS emissions.

[Figure]

*Figure S1. Modelled BC and measured eBC concentrations at the urban site during both campaigns (panel A), and modelled sources (panel B) using CAMS emission data and ADCHEM. Pie charts show the sources at the rural station during the respective campaigns.*

The figure shows that the eBC concentrations in the background air did not vary much between the two campaigns, although the sources are slightly different. Hence, the background size distribution could also have changed. However, to us, the diel variation of GMD in figure 3 convincingly shows that the BC core size is driven by local traffic emissions and not seasonal change. The nighttime rBC GMD during the urban campaign is very similar to the GMD duringthe rural campaign.

Changes to the manuscript:

*L62. Figure S1 shows that the BC concentrations at the rural site during July-October did not change drastically between the times of the two campaigns.*

*L193. The long-distance transported BC source contributions during the rural and urban measurement campaigns were estimated using the Lagrangian 1D-column chemistry transport model ADCHEM (Roldin et al., 2011; Roldin et al., 2019). ADCHEM was setup and run forward in time along pre-computed 14-days backward HYSPLIT air mass trajectories arriving 100 m a.g.l. at Hyltemossa, one new trajectory every hour. Source specific anthropogenic BC emissions along the trajectories were taken from the CAMS-GLOB-ANT v4.2 global emission inventory, which has a spatial resolution of 0.1°x0.1° (Granier et al., 2019). BC emissions from wildfires were estimated using the GFED4 emission inventory (Randerson et al., 2018).*

*Granier, C., Darras, S., Denier van der Gon, H., Doubalova, J., Elguindi, N., Galle, B., Gauss, M., Guevara, M., Jalkanen, J.-P., Kuenen, J., Liousse, C., Quack, B., Simpson, D., Sindelarova, K., The Copernicus Atmosphere Monitoring Service global and regional emissions (April 2019 version), Copernicus Atmosphere Monitoring Service (CAMS) report, doi:10.24380/d0bn-kx16, 2019.*

*Roldin, P., Swietlicki, E., Schurgers, G., Arneth, A., Lehtinen, K. E. J., Boy, M., and Kulmala, M.: Development and evaluation of the aerosol dynamics and gas phase chemistry model ADCHEM, Atmos. Chem. Phys., 11, 5867–5896, https://doi.org/10.5194/acp-11-5867-2011, 2011.*

*Roldin, P., Ehn, M., Kurtén, T., Olenius, T., Rissanen, M. P., Sarnela, N., Elm, J., Rantala, P., Hao, L., Hyttinen, N., Heikkinen, L., Worsnop, D. R., Pichelstorfer, L., Xavier, C., Clusius, P., Öström, E., Petäjä, T., Kulmala, M., Vehkamäki, H., Virtanen, A., Riipinen, I., and Boy, M.: The role of highly oxygenated organic molecules in the Boreal aerosol-cloud-climate system, Nat. Commun., 10, 4370, https://doi.org/10.1038/s41467-019-12338-8, 2019.*

*Randerson, J.T., van der Werf, G.R., Giglio, L., Collatz, G.J., and Kasibhatl, P.S., Global Fire Emissions Database, Version 4.1 (GFEDv4). ORNL DAAC, Oak Ridge, Tennessee, USA. https://doi.org/10.3334/ORNLDAAC/1293, 2018.*

*L258. Figure S1 shows that while BC concentrations at the rural site were similar during the two campaigns, the source sectors changed slightly. Notably, more industrial emissions and less shipping emissions affected the concentrations during the urban campaign.*

**Technically speaking there are some additional soft spots. It is very much not clear how the absorption data were treated and corrected, as a consequence the data are quite questionable. See specific comments. Moreover, the Aethalometer data are not essential to the scope of the paper since the SP2 provides mass concentration, size distribution and mixing state proxy.**

See answers for specific comments below to the technical questions. The aethalometer data is essential to strengthen our view on the importance of long range transport since we used a much longer dataset. Also, we think that the discrepancy between rBC and eBC measurements is of interest to the community, and should be investigated further.

**It must be described more clearly when ACTRIS data (absorption and chemical composition) were used. The reader realizes only towards the end of the manuscript that a year-long dataset was used. No description or introduction to these data is ever given.**

We have clarified what data was used with the following graphics and comments:

*L77. Figure 1 shows during which time the urban and rural campaigns took place and which data was used. BC properties and concentrations was determined using a Single Particle Soot Photometer (SP2) that was moved between the sites and Aethalometers (AE33) deployed permanently. Aethalometer data for the whole year was used from the rural site to assess source regions.*

[Figure]

*Figure 1. Overview of the data and instruments used.*

**SPECIFIC COMMENTS**

**L17: health effect is mentioned also in the introduction, but it is never really explained.**

This is beyond the scope of this paper, and we refer the readers to the cited literature.

**L34 It is an odd way to start a paper. I would remove the first sentence since it might apply to all atmospheric species.**

Although a platitude to aerosol scientists, it is our way of saying that soot is ubiquitous, which is a common way to introduce and motivate research on BC.

**L36: this is the motivation of your work and also the first sentence of the abstract. Still, I have zero ideas about how BC affects the climate and human health.**

The statements have references to back it up. We have added WHO global air quality guidelines 2021 as a reference to further strengthen the claims with recent findings.

**L40-41: I would provide an example for all properties or not report any example. Listing only the lensing effect does not add any relevant information since absorption enhancement is not a topic of the paper.**

Agree. We removed the sentence.

**L46-47: it appears like the reference for transport and deposited dose is missing.**

The sentence now reads (with an added reference for experimental deposition of diesel exhaust particles):

*L45. BC from different sources have different size distributions (Schwarz et al., 2008; Laborde et al., 2013; Saarikoski et al., 2021), that affects both transport (lifetime) and light interactions (Hinds, 2012), as well as deposited dose (Alfoldy et al., 2009; Rissler et al. 2012).*

*Rissler, J., E. Swietlicki, A. Bengtsson, C. Boman, J. Pagels, T. Sandström, A. Blomberg and J. Löndahl: Experimental determination of deposition of diesel exhaust particles in the human respiratory tract, Journal of Aerosol Science, 48, 18-33, 2012.*

**L50: Health and toxicity are mentioned a couple of times, but it is not yet explained how fundamental properties are connected to health.**

We refer the interested readers to plenty of well cited literature e.g. WHO, 2021; WHO, 2013; IARC, 2014.

**L55-56: main take-home message is summarized here. I think it does not belong to the introduction. Potentially makes more sense as a final statement of the abstract.**

We agree and moved this sentence to the abstract.

**L60-61: Please provide some evidence supporting your statement. July September is a long-time gap. I expect different dilution due to boundary layer height (I imagine lower temperature in October), different precipitation and washout, or different chemistry due to shorter sunlight duration…**

See answer and graphic to major comment regarding figures 1,2 and 3 above.

**L71-72: SP2 is not yet introduced. Move this last sentence to the instrumentation section. Since you mention turbulent flux. Can you estimate a loss fraction?**

We want to keep the description of inlets in this section so we changed the sentence to:

*L72. The single particle soot measurements at the rural site were conducted through a ¼'' tube at 20 liters per minute (turbulent flow), due to simultaneous Eddy Covariance flux measurements (not described here).*

The loss estimations are described in Section 3.2 and shown in the supplement.

**L74: This chapter is extremely long and complex. I would add a table listing the deployed instruments and measured variables at the two sites.**

We agree and have added the graphic above showing the different instruments and when they were used as well as subsections for the various instruments.

**L80: is good practice to provide the name, city and country of the manufacturer. Missing everywhere in the manuscript.**

We do not believe ACP has this as a house standard rule. We prefer without.

**L87: is this the saving rate of the SP2? A person not familiar with the SP2 would not understand what was does it mean. I think is not so important to be mentioned.**

Typically saving raw data signals from all particles gives too much data to handle by the analysis program. This does not affect the concentration estimations but is important for at what timescale it is feasible to extract a size distribution. We'd like to keep this description so that SP2 users knows the settings used.

**L90-91: What do you mean by "not satisfactory"? Was the reason connected to a wrong sizing of the DMA or a decrease in the performances of the SP2 (decreasing laser power, misalignment)? Are then the data valid?**

The calibration data of the rural campaign is missing due to a savings setting (human error). The instrument was calibrated before and after the campaign with similar results. Although this is definitely not ideal we strongly believe that the rBC data is valid. The instrument settings and alignments prior to each campaign ensures similar LII signals for a given mass. For instance, if the laser deteriorates for any reason, the current is typically amplified so that the power read by the output coupler stays constant. We changed a sentence to clarify:

*L100. The response of the broadband incandescence detector from typical ambient BC particle masses varied with less than ±15 % for calibrations performed before and after the rural campaign.*

**L98: 62 nm is a very small diameter, but those particles will not contribute significantly to the total mass. My issue with this choice is that most, if not the totality, of previous SP2 paper, reports rBC particles**

**starting from 80-90 nm…which I might consider the safe side. Considering that you do not provide any counting efficiency, I do not understand why you chose such a low cut-off.**

The cut off was chosen during analysis by setting the level of quantification threshold of the incandescence channel to the point where there's a sharp decrease in concentration. See figure below for an example. Since we don't make use of the number size distribution modes it doesn't matter for those results. For the number concentrations and fractions we discuss the limitations of the instrument and include the particles that we are certain are real. From personal experience with our SP2 as well as from intercomparison with other groups a D50 between 60-70 nm in mass equivalent diameter is quite normal.

[Figure]

**L104: the coating is not directly measured by these two detectors. I would use caution with these statements since people might think that the SP2 directly provides coating thickness of BC-containing particles, which is well far from reality.**

Agree. Changed to:

*L112. BC particle coatings can be estimated by the SP2 using a separate scattering detector.*

**L106-111: I am genuinely confused by this explanation. Are talking of delay-time or LEO-fit. To me, it appears like a mixture of the two. Please rewrite it. If the position-sensitive detector was not used is not worth mentioning, since it adds confusion.**

It is not easy to explain this without going into a great deal of detail, that is outside the scope of the manuscript. Instead we refer the reader to the cited literature. But we do say that we use the delay-time method and there is no mention of LEO or split-detector in the manuscript, so we don't know why there is confusion.

**L114: since you mention Weingartner…what Cref value was used? Was it calculated for this specific aethalometer or taken from Weingartner or Collaud-Coen or Zanatta? These papers are based on AE31 though and not AE33.**

Cref was not changed from the default value of 1.39 for the filters used. We added the following parenthesis:

*L123. Default corrections for filter scattering (Cref = 1.39) and loading effects were used (Weingartner et al., 2003).*

**L125-133: Please add the chemical species identified by the SP-AMS. If all presented rBC mass is derived from the SP2 the calibration for refractory material is even described? In what sense the SP-AMS at the urban site did not provide robust results (instrument malfunction, wrong calibration)? Is this the reason why all AMS graphs at the urban site are plotted with arbitrary units? I have the feeling that this issue and the SP2 calibration problem (L90) undermine the credibility (accuracy, reproducibility) of the dataset and, as a consequence, the full manuscript. The authors should explain in more detail why and how the SP2 and SP-AMS data are still reliable despite the technical issues.**

We added the species in the following sentence:

*L133. The (non-refractory) components of particles in the air beam are vaporized on a heated tungsten plate (600 °C), subjected to 70 eV electron ionization, and the ions produced are detected and categorized (organic, nitrate, sulfate, ammonium or chloride) by a high-resolution time-of-flight mass spectrometer.*

We also removed the description of rBC calibration of the SP-AMS.

For the SP-AMS, there was confusion around the measured mass loadings (based on field calibration with 300 nm mobility diameter ammonium nitrate particles). The rural loadings were higher throughout the campaign, despite very similar time trends for all non-refractory species (see Figure 7). We found the concentration in the rural data well supported by DMPS (same site) and FIDAS (nearby background site). We do not have measurements for a similar evaluation of the urban SPAMS. Hence, the combined evidence suggests the urban absolute concentrations from SPAMS are too low, which we attribute to calibration issues. Notably this would not have been detected if we were not comparing with another SPAMS downwind (at the rural site) which was properly supported by auxiliary measurements.

That the SP2 data is valid we hope is convincing by the answer to comment on L90-91 above. For the SP-AMS data, we simply use the fractions of chemical species and not the absolute mass concentrations in our interpretations.

**L145: just a comment to point out that 64 nm of electric-mobility diameter does not correspond to 64 nm of mass equivalent diameter.**

True. We removed that part in the methods section. We choose this value as a conservative minimum. It is tricky to compare the two instruments without a common size selection. In the results we also compare the number fractions using a larger diameter, to give the reader a range.

**L171: back trajectories are not shown**

Showing an example trajectory would not add to the manuscript.

**RESULTS AND DISCUSSION**

**L183-191: There is no context to your observation. Up to me, these are low concentration for being an urban site.**

It is true that the concentrations in Malmö are not that high compared to many other cities around the world. However, comparing Malmö to other sites and a wider context using only this short-term campaign would not suffice. We want to compare the nearby urban and rural sites only.

**L186: it is already clear from its concentration that the curbside is not extremely polluted. At what percentage difference you would define extreme pollution? And based on what process?**

True, we compared the two sites to show that the curbside was representative of the city. We removed the word exreme.

*L209. This suggests that the curbside measurements are indicative for the city.*

**F1: provide error bars. Does the analysis include the weekends?**

We added error bars, see new figure below. Yes, weekends are included.

[Figure]

**L191: MAC of BC is not the only reason for the difference in AAE. Different AAE might be caused by a change in the chemical composition of absorbing aerosol or a change in the relative concentration of aerosol absorbing more light at the lower wavelength. If this is the case, MACbc will remain the same while AAE of total aerosol will increase.**

True. We rephrased the sentence:

*L214. The AAE is similar between the sites, with small differences likely owing to different BC sources.*

**L200-212: As it is also pointed out in the text, the rBC number fraction mostly depends on the diameter quantification limits of both the SP2 and especially DMA rather than on aerosol properties. So, I am not sure what should I retain out of this subchapter.**

We agree that these are not hard numbers, but still think it is important. The lower limit is definitely too low since fresh BC particles without coating would have a larger mobility diameter than its mass equivalent diameter and the SP2 may not have 100% counting efficiency. The larger limit assumes that all BC particles counted by the SP2 are at least 130 nm in mobility diameter which is likely closer to the truth given the structure of fresh soot agglomerates. However, collapsed particles with little coating might be unaccounted for by this higher limit. The limits we choose give an estimate of BC number fractions between 2.7-6 % and 13.4-45 % for the rural and urban sites respectively. We think this difference is interesting and something that can be compared to in future studies.

**L213-223: Here several factors must be considered and I strongly believe that the SP2 size cut is not the problem. Even if the SP2 size range could be easily accounted for by fitting a lognormal to the size distribution. Anyhow, no details are reported about the correction used for the AE-33 and I believe the problems are more connected with absorption calculation rather than the SP2 detection range. I recalculated the MAC from the values reported in Table 1. I recalculated Babs by multiplying Mebc by 7.77 m2/g. Then I calculated MAC as the ratio of Babs and MrBC. So, I obtain MAC values above 25 m2/g. This value is very similar to the mass attenuation coefficient used in the past to convert directly attenuation coefficients to eBC in the old AE31. So, I cannot say what happened here exactly, but I think that no Cref or correction was applied. I think you need to do a bit of extra thinking here and reconsider the relevance and accuracy of eBC measurements.**

We used the default AE33 settings that is commonly used to report BC values and recommended by e.g. ACTRIS (https://www.actris-ecac.eu/particle-light-absorption.html). Since we report the MAC that was used anyone, as you did, can recalculate the Babs values. We do not want to use the SP2 rBC as a reference to report some equivalent rBC from the Aethalometer. As pointed out above, the correct Cref value was used. The difference between different BC instruments is ongoing research, e.g. ACTRIS now recommends a "Harmonization factor" to adjust AE33 Babs values. This a very important and interesting discussion and as the references in this paragraph show, we are not the first to report this difference.

**L2017: was the 10-20% calculated or is it just a gentle guess? If it is calculated why is not applied to the measurements? Lensing effect cannot be excluded, but I hardly think that this is the main reason behind the eBC-rBC difference: You are using a MAC 1.6 times smaller than Swedish ambient values (Martinsson), while an additional 10-20% is coming from size cut. There is too much uncertainty to speak about absorption amplification.**

This is an estimation from the example shown in figure S6. We agree that there is a lot of uncertainty in this discussion but again, it is important and interesting. The MAC calculated by Martinsson et al. used EC measurements. For example, Pileci et al. reported up to a factor of 2 difference between rBC and EC measurements from several European sites. We report the eBC values using standard Aethalometer settings of MAC. MAC values can certainly be site specific, but also have a seasonal dependence. Zanatta et al. calculated site specific values for a nearby site in southern Sweden that were even lower than the default values we used. We try to discuss potential reasons for the discrepancy but do not have a conclusive answer. We believe that this could be an interesting paper on its own, perhaps using longer datasets and/or lab campaigns.

Pileci, R.E., Modini, R.L., Bertò, M., Yuan, J., Corbin, J.C., Marinoni, A., Henzing, B., Moerman, M.M., Putaud, J.P., Spindler, G. and Wehner, B., 2021. Comparison of co-located refractory black carbon (rBC) and elemental carbon (EC) mass concentration measurements during field campaigns at several European sites. Atmospheric Measurement Techniques, 14(2), pp.1379-1403.

Zanatta, M., Gysel, M., Bukowiecki, N., Müller, T., Weingartner, E., Areskoug, H., Fiebig, M., Yttri, K.E., Mihalopoulos, N., Kouvarakis, G. and Beddows, D., 2016. A European aerosol phenomenology-5: Climatology of black carbon optical properties at 9 regional background sites across Europe. Atmospheric environment, 145, pp.346-364.

**L220-223: so why not use the Martinsson MAC? There is no explanation behind the choice of 7.77 m2/g.**

See discussion above. The default MAC is what is used by many to report BC.

**F2: what is the small window in the plot?**

Insert shows the normalized distribution for comparison of the modes (already in caption).

**L234: the fact that aerosol diameters are affected by cloud processing is true. I wonder how this is relevant to your study. If this is a general statement, please provide at least some references.**

We removed this statement to not add confusion. Now it reads:

*L256. The difference in mass GMD is statistically significant and can be explained by different BC particle sources, air masses and coagulation.*

**F3: legend should simply describe what is shown in the graph. Avoid adding interpretation of results, this belongs to the text. I am not sure what the crosses indicate. I imagine that whiskers are 10-90 percentile…missing**

The + is outliers, which is mentioned in the caption. The whiskers show the max and mean of data (as long as not outliers). This is standard for boxplots. We deleted the interpretation so caption now reads:

*Figure 4. Box-plot showing the daily pattern of rBC GMD (2 h averages) as measured by the SP2, at the urban and rural sites. Boxes show the median, 25th and 75th percentiles, with values more than 1.5 below or above those considered outliers (+).*

**L242: very few people know what is the broadband channel. Since it is not essential information to interpret your result, I would remove it in the result section.**

Agree, we removed broadband. Sentence now reads:

*L266. The fraction of "thickly coated" BC particles with an incandescence peak height corresponding to a BC core diameter of mass 2.4 fg (~136 nm mass equivalent diameter) was on average 71 ± 8 % (1σ) at the rural site and 38 ± 7 % (1σ) at the urban site, with no big difference during weekends.*

**L245: these effective density measurements are interesting. You could show the mass distribution for all selected diameters…it would help to understand your text. I might have missed this info, but did you measure effective density at the rural site? Do you see this bimodal distribution?**

No, the DMA-APM measurements were only performed in the city. As discussed in section 3.4 we choose to show the 150 nm run since that is where the bimodality was most clear (which is a balance between the fraction of soot particles and total number of particles to get good statistics). Below we show some examples of other sizes (from left to righ: 75 nm, 100 nm, 250 nm). We already included the results of other sizes in the manuscript but don't think showing the other sizes would add to the manuscript main points.

[Figure]

**L271: At what altitude the back trajectories are passing over Malmo?**

50 m above ground level. Added the following to the manuscript:

*L295. From time series comparison with HYSPLIT trajectories, it could be seen that some peaks in eBC, non-refractory PM1, and total particle number concentrations coincide with trajectories passing 50 m above ground level over Malmö/Copenhagen before arrival at the rural site.*

**F4: what is this arbitrary unit?**

Mass concentration. Figure changed:

[Figure]

**L275: No actual description is given for the one-year-long dataset. It is very confusing, especially without a dedicated subsection with numbering and title.**

We hope that the addition of subsections and graphics above have helped clarify this.

**L300-320: did you use the one-year-long dataset for the malmo influence? Not clear to me. Specify the use of the 1year dataset. If not, I am not very persuaded that your 15 days measurements could show or not show any systematic impact of Malmo emission on background aerosol concentration.**

We agree that 15 days is short and this is exactly why we also used the one year eBC dataset to further investigate the Malmö/Copenhagen influence. See answers above regarding clarifications.

**L331: ok, but why?**

See discussion regarding L213 and L217 above.

**L339: first-time HR-ToF-AMS is mentioned**

We have changed to SP-AMS

**L349-352: what are regional traffic sources? You clearly show that the urban environment is impacted by traffic; so, emission reduction policies will benefit the urban population. But it will have a less evident impact on rural locations. I find this final statement a bit confusing, I suggest rephrasing.**

We agree that this reads a bit confusing and have rephrased the last paragraph to the following:

*L375. The results show that local urban emissions of BC in southern Sweden have a small effect on concentrations and properties in the regional background air. Local abatement strategies aimed towards reducing BC emissions from traffic sources will thus have an effect mostly limited to the urban population. The contribution of long-range transported BC will be even more evident when diesel-driven vehicles are now rapidly being replaced with other forms of propulsion, such as electric motors or hydrogen fuels.*

Furthermore we found an erroneous use of the word regional in the results section. L293 in the submitted manuscript was changed from:

L293. It is clear that non refractory-PM1 concentration and composition are driven by regional pollution at both sites, while black carbon is more influenced by local emissions. Regional background sources dominates even the organic aerosol, for which one could expect a larger difference between the sites owing to the high urban traffic density (Glasius et al., 2011).

To:

*L318. It is clear that non refractory-PM1 concentration and composition are driven by long-range pollution at both sites, while black carbon is more influenced by local emissions. Long-range sources dominates even the organic aerosol, for which one could expect a larger difference between the sites owing to the high urban traffic density (Glasius et al., 2011).*

**REVIEWER #2**

**The article of Ahlberg et al. investigated black carbon concentrations, size distributions, mixing state and sources in two measurement campaigns in Southern Sweden; urban and rural locations. They measured BC by a single particle soot photometer and aethalometer, and complemented the measurements by number size distribution and PM1 chemical composition measurements. In addition to examine the characteristics of BC, the aim of this study was to investigate the contribution of regional sources and long-range transport to the observed BC concentrations.**

**BC concentrations were larger at the urban site than at the rural site, especially during the traffic rush hour. Also the number of particles having BC core was larger at the urban site and BC particles were smaller in size and had thinner coating. Based on the trajectory analysis, air masses coming from Eastern Europe comprised twice as much BC compared to Western Europe. Nearby Malmö/Copenhagen region impacted rural site BC concentrations only to some extent.**

**This article presented new results from the field measurements and had enough conclusions for a measurement report. However, the paper is missing a lot of details (for example the measurement sites and periods) and is therefore partly unclear for the reader. This paper merits publication in ACP measurement reports after addressing the comments given below.**

**General comments**

**The paper is confusing for the reader as there are so many different measurement periods that are not clearly described in text. For example, the rooftop urban site is only mentioned in one sentence even though the results from the rooftop are shown in supplemental material and discusses in text. Also, the one year measurements for BC are not mentioned in methods section at all. A table or figure listing all sites, instruments and measurement periods would help the reader to get an overall picture of the dataset utilized in the paper.**

We hope that the implemented changes discussed above have helped the readers. Specifically the addition of subsections and  graphic showing the data used. We did not include the rooftop measurements in this graphic since it was only used to show that the curbside measurements are valid as an indicator of urban BC as discussed in section 3.1 and shown in figure S4.

**Specific comments**

**Title:  The title does not cover all the aspects of the text. Based on the title, the study focuses on background air and the effect of regional sources, whereas to me, the object of this paper is much broader. I suggest considering to change the title to reflect better the whole study.**

We have tried to change the title into something more suiting. Our new title is "Black carbon properties and concentrations in Southern Sweden urban and rural air – The importance of long-range transport".

**Abstract**

**lines 22-23: …. higher than the background levels… higher than background levels at the urban site or at the rural site? Specify**

Clarified that it is compared to rural background (and stitched together with the previous sentence):

L21. Equivalent and refractory BC mass concentrations at the urban site were on average a factor 2.2 and 2.5, with peaks during rush hour up to a factor ~4, higher than the rural background levels.

**line 25: fresh plumes of traffic?**

Yes, clarified.

*L24. The organic components of the fresh traffic plumes were similar in mass spectral signature to "hydrocarbon-like organic aerosol" (HOA), commonly associated with traffic.*

**line 25: "hydrocarbon-like organic aerosol", why parentheses?**

This is a common acronym, especially for AMS-users.

**Methods**

**lines 64 and 70: PM cut-off was different at urban and rural site (PM2.5 vs PM10). How much that impacted the BC results, can you estimate?**

Good point. The bias that could have been introduced is that the measured BC concentrations at the urban site was too low. We don't believe a lot of BC mass is present in the larger particles. Especially the locally emitted particles will not have had time to coagulate and grow. Viidanoja et al. (2002) showed that typically more than 90% of BC resided in PM2.5 at an urban site in Finland. Either way, it is highly unlikely that this would affect our conclusions.

Viidanoja, J., Sillanpää, M., Laakia, J., Kerminen, V.M., Hillamo, R., Aarnio, P. and Koskentalo, T., 2002. Organic and black carbon in PM2. 5 and PM10: 1 year of data from an urban site in Helsinki, Finland. Atmospheric Environment, 36(19), pp.3183-3193.

**lines 64-65: add details of rooftop measurements**

The rooftop measurements are the official Malmö municipality urban background sampling site. We included some details:

*L66. The validity of the curbside site measurements as an indicator of the BC concentrations in the city was assessed by comparing simultaneously collected data at a rooftop urban background site (20 m a.g., PM2.5 inlet).*

**lines 71-72: Why SP2 sampling system is described here only for the rural site?**

At the rural site the SP2 sampled through an inlet different from the other instruments. At the urban station all instruments were connected to the described inlet.

**line 76: "During the latter part of the urban campaign, chemically resolved particle constituents were measured simultaneously at both sites…" were there simultaneous measurements only for the SP-AMS? Also for AE?**

Yes, as can be seen in figure 8. We have clarified that we have used the rural Aethalometer data for the full year in the discussion above.

**lines 128-129: It's very difficult to understand why the mass concentrations from the SP-AMS could not be calculated at the urban site as the SMPS number size distributions can be converted to mass size distributions. Were there some issues with the SPMS data as well?**

This is also a good point and we tried comparing with both SMPS and TEOM data. For the SMPS, the lower cut-off size meant that the two instruments likely did not see the same mass, and that the mass-size distribution was often cut in the middle of a mode, which introduces a lot of temporal variability. For the TEOM data (not discussed in the manuscript), there was a separate issue with an erroneous baseline. We are still comfortable using the chemical speciation, but not the absolute mass concentrations.

**lines 133-139: traffic plumes, did you have any gas monitors (NOx, CO2, CO) that could have been used to indicate traffic plumes as well? How was the diurnal distribution of traffic plumes, were they detected only during rush hours?**

There are gas monitors permanently deployed at the urban monitoring site, but we did not look further into this data since plumes were easily discernible using both AMS and SP2. The selected plumes were both from rush hour and night-time.

**line 157: APM was run only during five days in spring 2019; how comparable is this data to summer/autumn 2018 data in terms of weather and traffic volume/fleet?**

Traffic is rather similar throughout the year, except perhaps during vacation times. Those measurements, to verify the bimodal structure of the aerosol effective density, are just a snapshot that agrees well with the Rissler et al. values.

**lines 168-170: why the trajectory analysis was carried out only for the rural site, why not both sites?**

Our aim was to investigate the origin of BC in the rural background air and specifically if air passing over Malmö had a significant impact. Hence we only used the rural dataset.

**Results and discussion**

**line 184-185: traffic intensity; any figure on that?**

Below is a figure showing number of cars (orange, left axis) and $CO_2$ (blue, right axis) at the site. Data comes from Malmö municipality. We believe this is quite typical behaviour, and not needed in the manuscript.

[Figure]

**line 192: "The AAE is similar between the sites with small differences…" To me, 1.13 and 1.24 are not that similar. Could you speculate more the reasons for the difference? Also, AAE is smaller at traffic site than at rural site, is this typical trend?**

AAE is actually higher at the traffic site, which may be surprising given the larger contribution of fresh traffic aerosol compared to the rural site. The numbers are significantly different (95% CI), but we don't want to add too much speculation on the reasons, since it concerns short term campaigns that were not simultaneous.

**lines 244-257: bimodal distribution, could you discuss more on the sources of two modes?**

The two modes comprise of fresh agglomerated soot with a density lower than 1 $g/cm^3$. The other mode contains more spherical particles (all except soot) which are likely aged. We refer the reader to Rissler et al. (2014) for further reading on the effective density and mixing state of urban aerosol.

**lines 270-271: "It was clear that SW winds had more occurrences of high eBC than e.g. NW winds." I somewhat disagree with this sentence. It is clear that the occurrence of SW winds was higher but eBC concentrations are difficult to compare based on Fig. 5 since the occurrence of NW winds is so low. However, Fig 6. shows nicely larger concentration of eBC related to the SE winds.**

We agree, it is hard to see in that figure. We removed that sentence.

**lines 314-315, correlation of non-refractory-PM1 and thickly coated BC fraction (Fig 8); why 24-hour averaged data with only 15 data points? Why not for example 1-hour averaged data? Was it because of SP2 data?**

Yes, the thin/thick coating fraction was evaluated on a 24h basis. In theory it can be done with a higher time-resolution, but this was not implemented in the version of the analysis tool we used.

**line 317: non-refractory-PM1 (dominated by secondary material); HOA related to traffic plumes was discussed earlier but there is no data showing that organics were mostly secondary. Could you add some contributions for primary (HOA?) and secondary OA?**

In order to quantify our statement that OA at the urban site was dominated by OOA, we used the "m/z tracer method", colloquially known as "poor person's PMF", from Ng et al 2011 stated to reproduce PMF results within 30%. Our result is 78% OOA, 9% HOA for the campaign averaged mass spectrum.

We added the following to the methods:

*L150. The m/z tracer method (Ng et al., 2011) was used to estimate the different OA components.*

And the following to the results:

*L320. The organic aerosol at the urban site was clearly dominated by oxygenated organic aerosol (OOA, 78%) with only a minor contribution of HOA (9%), suggesting that aged long-range sources dominate even the organic aerosol, for which one could expect a larger difference between the sites owing to the high urban traffic density (Glasius et al., 2011).*

Ng, N.L., Canagaratna, M.R., Jimenez, J.L., Zhang, Q., Ulbrich, I.M. and Worsnop, D.R., 2011. Real-time methods for estimating organic component mass concentrations from aerosol mass spectrometer data. Environmental science & technology, 45(3), pp.910-916.

**Summary and conclusions**

**lines 341-342 "…but composed of a small fraction of the total aerosol." How much?**

9%, see answer right above.

**Supplemental material**

**Fig. S4: Add instruments and time-resolution of data**

This is now figure S5. The caption was updated:

*Figure S5. Correlation between 1h averages of Aethalometer (eBC) and SP2 (rBC) data.*

**Technical corrections**

**line 185: …levels show are similar… correct**

Corrected:

*L207. The comparison between the urban street-level and urban background eBC levels are similar in time-series but with a lower daily maximum for the roof-top measurements (Fig. S4).*

**line 339: change HR-ToF-AMS to SP-AMS**

Changed.

---

## Referee Report (RR1)

This manuscript (acp-2022-156) reports black carbon concentrations and physical properties in both urban and rural environments in Sweden. The study aims to investigate whether the rural air was influenced by traffic emissions and long-term aerosol transportation. I am glad to see that the authors have done various analyses using the available measurements (such as BC coating estimation, trajectory analysis, and OA source apportionment) and reported many observations. However, the Results and Discussion section missed many explanations, and the logistic of some subsections is not clear (See my general comments below). I suggest a major revision and restructuring of the paper before being acceptable.

**Major comments:**

1. I understand that the authors had to move the SP2 between the two campaigns, and some instruments were not available during some periods of the study. But this experimental limitation significantly hinders the authors' goal to explore the influence of urban emissions on the rural air quality, because the two campaigns were not conducted simultaneously. The authors use Figure S1a to justify that the BC concentrations at the rural site did not change drastically from July to October, but this result is not enough to support the research's goal because, besides BC, many other atmospheric components and meteorology can be changed between the two campaigns. I am not against the author's effort to compare the urban against the rural environments, but the author should be careful to use asynchronous observations to explore how urban emissions affect rural background air and draw the conclusion that "local abatement strategies aimed towards reducing BC emissions from traffic sources will thus have an effect in reducing the BC mostly limited to the urban population." Overall, I strongly suggest revising the goal and the conclusion.

2. Section 3.6 (origins of BC in the rural background air) is not well structured. The authors presented a lot about the origins of aerosols at the urban site (lines 331 -360), but didn't really answer why the BC is from at the rural site. It seems that the answer is related to Figure 7, but the paragraph above Figure 7 is hard to understand. The authors should consider not using a histogram to present Figure 7, then rephrase the paragraph to better justify how wind directions and precipitation affect the rural background air.

3. The introduction section should include more details. First of all, the authors should add references to the statements in the first paragraph. Second, the second paragraph is not directly related to the research question. The paragraph is a very general description of BC measurements and properties. More details should be added, including how BC properties vary from different emission sources, how BC properties change after mixing with other organic compounds, and what the authors mean by the importance of the BC mixing state. Third, the authors should add another paragraph to introduce the

discrepancies between eBC and refractory BC measurements since the authors present such results in the Result and Discussion. Lastly, if the author wants to keep using the current title, an overview of the influence of long-range transportation of aerosols on BC properties from the other studies is necessary.

**Specific comments:**

1. Line 39: The authors should specify what kind of severe climate and public health effects can be introduced by BC.
2. Line 66: Describe what the two campaigns are. Actually, the rural site, rural campaign, urban site, urban campaign are very confusing. I suggest renaming the rural campaign and urban campaign. Maybe just Campaign #1 and #2.
3. Line 67: Is the results in Figure S1 for the rural or the urban site? The main text says rural, but the caption of Figure S1 says urban.
4. Line 71 and Line 78: The two sites either measures PM2.5 or PM10. Does this affect any results?
5. Line 101: Was 10 and 1-5% of the data discarded or used in the analysis? Please specify.
6. Line 151: The authors should consider adding a time-series figure of BC and $C_4H_9^+$ concentrations during a traffic plume, then label the three windows and their durations.
7. Line 186: I don't get how the authors concluded that "these corrected values are closer to the true values". What do the authors mean about "true values"? Please explain in the main text.
8. Line 216: Why does the greater concentration measured at the curbside suggest that these measurements are indicative of the city? Please explain in the main text.
9. Line 266: Show statistical results to justify "statistically significant".
10. Lines 275-293: Do the authors have any explanation about why 50 and 75 nm particles are unimodal, but 100 and 150 nm particles are bimodal?
11. Line 307: I don't find the results of HYSPLIT trajectories in the main text and SI.
12. Line 340: How do the authors draw the statement that "approximately half of the mass at the urban site is due to long-range transport"? First, how "half of the mass" is estimated? Second, what is the evidence for "long-range transport" in Figure 8? Please explain this sentence in more detail in the main text.
13. Figure 8: Why does panel d have a different y-axis label from panel c?
14. Line 353: The authors showed the strong correlation between non-refractory PM1 concentration and thickly coated BC. What is the correlation between non-refractory PM1 concentration and non-coated BC (i.e., fresh BC)? If the latter correlation is also strong, the statement "the BC coatings are similar in composition to non-refractory-PM1" is over-interpreted.

15. Line 368: I don't think a factor of 2.2-4 between urban and rural environments is "surprisingly low". The authors should consider using another word.
16. Line 388: As I mentioned in the major comment #1, the authors should rephrase this sentence.

---

## Referee Report (RR2)

The authors of the manuscript (acp-2022-156) have well addressed most of my comments in the last round of revision. However, there remain two comments that I highly recommend the authors consider addressing before publication. Once the comments are addressed, in my opinion, the manuscript is suitable to be accepted by the journal.

1. I still think the current introduction section is too general; instead, it should include more explanation about the variabilities of soot from different sources (the first paragraph). This is very important to the readers to understand why the authors focus on comparing absorption, size, and mixing state of BC between the sites.

2. Add a time-series figure of BC and C4H9+ concentrations: The authors agreed that this could be a nice analysis, but decided not to add it to the current manuscript. Please add such a figure or explain why not adding. The figures can help readers understand the temporal variability of BC and HOA during a traffic plume. Adding the figure to the SI is fine.

---

## Author Response (AR2)

We thank all reviewers for their comments. Our answers, in normal text, are presented below the comments, which are bold. Changes to the manuscript are in italic with line numbers of the modified manuscript.

**REVIEWER #1**

**This manuscript (acp-2022-156) reports black carbon concentrations and physical properties in both urban and rural environments in Sweden. The study aims to investigate whether the rural air was influenced by traffic emissions and long-term aerosol transportation. I am glad to see that the authors have done various analyses using the available measurements (such as BC coating estimation, trajectory analysis, and OA source apportionment) and reported many observations. However, the Results and Discussion section missed many explanations, and the logistic of some subsections is not clear (See my general comments below). I suggest a major revision and restructuring of the paper before being acceptable.**

**Major comments:**

**1. I understand that the authors had to move the SP2 between the two campaigns, and some instruments were not available during some periods of the study. But this experimental limitation significantly hinders the authors' goal to explore the influence of urban emissions on the rural air quality, because the two campaigns were not conducted simultaneously. The authors use Figure S1a to justify that the BC concentrations at the rural site did not change drastically from July to October, but this result is not enough to support the research's goal because, besides BC, many other atmospheric components and meteorology can be changed between the two campaigns. I am not against the author's effort to compare the urban against the rural environments, but the author should be careful to use asynchronous observations to explore how urban emissions affect rural background air and draw the conclusion that "local abatement strategies aimed towards reducing BC emissions from traffic sources will thus have an effect in reducing the BC mostly limited to the urban population." Overall, I strongly suggest revising the goal and the conclusion.**

The aim of the study was to compare the BC properties between the sites, and to look for influence of the urban are on the rural background air. We believe that it has been clearly shown that the difference in BC properties is driven by the local traffic emissions in the urban area (Figures 2, 3 and 4). When it comes to the influence of the urban area on the rural background air, single plumes were hard to distinguish during the campaigns and therefore we expanded the analysis to the full year using available eBC and trajectory data (Figure 7), hence the cited conclusion in the comment is NOT from asynchronous observations. However we agree that the last paragraph can be misinterpreted and removed that part from the conclusions and slightly rephrased the paragraph above to be more suitable as a final conclusion. It now reads as follows:

*L381. Plumes from the nearby urban site to the rural site were not clearly distinguishable during the field campaigns. Trajectory analysis of the full year of 2018 show that significant increases in eBC concentrations at the rural site for air masses passing over the Malmö/Copenhagen area are only clearly seen during days with low precipitation. This increase, however, is small in comparison with the influence of long-range transported BC. Transport of BC from continental, and especially eastern, Europe is what governs the BC concentrations in southern Sweden background air, when looking at eBC from the full year of 2018.*

**2. Section 3.6 (origins of BC in the rural background air) is not well structured. The authors presented a lot about the origins of aerosols at the urban site (lines 331 -360), but didn't really answer why the BC is from at the rural site. It seems that the answer is related to Figure 7, but the paragraph above Figure 7 is hard to understand. The authors should consider not using a histogram to present Figure 7, then rephrase the paragraph to better justify how wind directions and precipitation affect the rural background air.**

We have tried to simplify the formulations and refer to section 2.2.8 for details on the analysis, to make the first paragraph of section 3.6 more clear. Further, we also added the median numbers in the legend of Figure 7.

*L302. Figure 6 shows the wind direction probability together with eBC from the five weeks rural campaign. The Malmö/Copenhagen area is in the direction 200 – 230 degrees as seen from the rural site. Based on the HYSPLIT trajectory analysis (Sect. 2.2.8) , some peaks in eBC, non-refractory PM1, and total particle number concentrations at the rural site coincide with air masses originating over Malmö/Copenhagen. These urban plumes were discernible when the air mass was relatively clean (typically of western or north-western origin). However, no conclusive results on the influence of the urban site on rural levels could be derived from analysis of the limited field campaign data. Also when analyzing the complete eBC dataset from 2018 there is no significant difference in the median eBC concentration for air masses with and without influence from Copenhagen and/or Malmö, according to the trajectory analysis. However, if we only consider air masses with insignificant precipitation within 48 hours upwind the rural site (i.e. less than <1 mm precipitation according the HYSPLIT model), the measured eBC median concentration is significantly higher in the air masses that have moved over Copenhagen/Malmö, 354.7 ng m-3 compared to 226.9 ng m-3 without influence from Copenhagen/Malmö. Similar results are also found when only the air masses originating from SW is considered (see Fig. 7 and Tables S2-S4). Most likely the contribution from the BC emissions from Copenhagen/Malmö is not apparent when analyzing the whole eBC dataset from*

*2018 because the SW air masses are generally influenced by more precipitation (more BC wet scavenging) compared to other air masses. The HYSPLIT trajectory simulations for the complete year show that 66 % of all air masses that passed over Malmö/Copenhagen were influenced by >1 mm precipitation within 48 hours upwind the rural site, compared to 39 % for all other air masses. The lowest median eBC concentration is found in air masses from NW (101 ng m-3) followed by NE air masses (149.9 ng m-3) and SW air masses (192.0 ng m-3). SE air masses clearly stand out from any other air masses with a median eBC concentration of 563 ng m-3. This can partly be explained by the low probability of precipitation in SE air masses. Only 23 % of these air masses are influenced by >1 mm precipitation within 48 hours upwind the rural site. However, also when considering the effect of precipitation upwind Hyltemossa the SE air masses have a median eBC concentration that is ~2 times larger than the SW air masses.*

[Figure]

*Figure 7. Trajectory analysis of different air mass origins, with corresponding median eBC mass concentrations in the legend. Panel A shows histograms with the observed eBC from the Aethalometer at Hyltemossa from year 2018 for air masses originating from SE, NE, NW and SW as defined in Fig. S3. Panel B shows the eBC histogram from all SW air masses with or without influence from the Copenhagen and Malmö region.*

**3. The introduction section should include more details. First of all, the authors should add references to the statements in the first paragraph. Second, the second paragraph is not directly related to the research question. The paragraph is a very general description of BC measurements and properties. More details should be added, including how BC properties vary from different emission sources, how BC properties change after mixing with other organic compounds, and what the authors mean by the importance of the BC mixing state. Third, the authors should add another paragraph to introduce the discrepancies between eBC and refractory BC measurements since the authors present such results in the Result and Discussion. Lastly, if the author wants to keep using the current title, an overview of the influence of long-range transportation of aerosols onBC properties from the other studies is necessary.**

We added references to the last sentence:

L39. *Soot measurements are complicated by the fact that the ambient aerosol is a dynamic mixture, and that soot from different sources may not possess the same properties in terms of chemical content (Malmborg et al., 2019), light absorption (Sandradewi et al. 2008), size (Schwarz et al., 2008), and toxicity (Hakkarainen et al., 2022).*

The second paragraph discusses the importance measuring not only absorption but size and mixing state of BC. This is in our opinion highly relevant properties that we want to compare between the two sites. The sentence on mixing state says that it is important to know to understand the light absorption. We added some details on BC transformation in the atmosphere. The paragraph now reads:

L43. *Although BC mass concentrations are routinely estimated from optical methods, BC number concentrations, size distributions and mixing state are rarely measured. In a previous study it was shown that global aerosol microphysics models underestimate the BC particle size, by a factor of ~2-3, while overestimating the number concentrations, by more than a factor of 3, compared to airborne measurements using single particle laser induced incandescence (Reddington et al., 2013). BC from different sources have different size distributions (Schwarz et al., 2008; Laborde et al., 2013; Saarikoski et al., 2021), that affects both transport (lifetime) and light interactions (Hinds, 2012), as well as deposited dose (Alfoldy et al., 2009; Rissler et al., 2012). Upon ageing in the atmosphere, fresh BC particles will obtain a coating layer consisting of*

*condensed materials changing both their hygroscopicity (Swietlicki et al., 2008) and light absorption properties (e.g. Zhang et al., 2018). This transformation depends on atmospheric conditions and constituents but can be on the order of hours under favourable conditions (Eriksson et al., 2017). Recent studies have pointed to the importance of BC mixing state, governed by emission source and atmospheric ageing, in understanding the light absorbing properties, and hence climatic impacts, of BC containing particles (e.g. Liu et al., 2017; Liu et al., 2019; Fierce et al., 2020; Yuan et al., 2020). These properties of BC, that are crucial to understand both health and climate impacts, are not measured by the BC measurement techniques commonly used by monitoring networks.*

The discussion on rBC and eBC is included in the Discussions section but we don't think it is necessary to include in the introduction.

**Specific comments:**

**1. Line 39: The authors should specify what kind of severe climate and public health effects can be introduced by BC.**

We prefer to refer the readers to the cited literature instead.

**2. Line 66: Describe what the two campaigns are. Actually, the rural site, rural campaign, urban site, urban campaign are very confusing. I suggest renaming therural campaign and urban campaign. Maybe just Campaign #1 and #2.**

We don't understand what is meant by "what the two campaigns are"? To satisfy another reviewer we added Figure 1 which shows an overview of the urban and rural measurements, when they were undertaken and what data was used. We don't think renaming the campaigns #1 and #2 would be less confusing, since we would still have to discuss urban and rural settings and sites.

**3. Line 67: Is the results in Figure S1 for the rural or the urban site? The main text says rural, but the caption of Figure S1 says urban.**

It's the rural site, thank you for noticing. We changed the caption.

**4. Line 71 and Line 78: The two sites either measures PM2.5 or PM10. Does this affect any results?**

This is our answer to a similar question in review round 1:

Good point. The bias that could have been introduced is that the measured BC concentrations at the urban site was too low. We don't believe a lot of BC mass is present in the larger particles. Especially the locally emitted particles will not have had time to coagulate and grow. Viidanoja et al. (2002) showed that typically more than 90% of BC resided in PM2.5 at an urban site in Finland. Either way, it is highly unlikely that this would affect our conclusions.

Viidanoja, J., Sillanpää, M., Laakia, J., Kerminen, V.M., Hillamo, R., Aarnio, P. and Koskentalo, T., 2002. Organic and black carbon in PM2. 5 and PM10: 1 year of data from an urban site in Helsinki, Finland. Atmospheric Environment, 36(19), pp.3183-3193.

**5. Line 101: Was 10 and 1-5% of the data discarded or used in the analysis? Please specify.**

That is what was saved. Changed to:

*L101. 10 and 1-5 % was saved in the rural and urban campaigns respectively*

**6. Line 151: The authors should consider adding a time-series figure of BC and C4H9+concentrations during a traffic plume, then label the three windows and their durations.**

We agree this could be a nice analysis, but feel it would not add to the current manuscript.

**7. Line 186: I don't get how the authors concluded that "these corrected values are closer to the true values". What do the authors mean about "true values"? Please explain in the main text.**

Figure S2 shows that the losses in the turbulent flow inlet is 20-30% for particles of diameters between 100-1000 nm. The SP2 number and mass size distribution peaks at below 100 nm and ~150 nm respectively, but that does not include coatings. Hence from the calculated losses, we estimated and corrected for 25 and 30% losses in Number and mass concentrations.

We changed the wording to not confuse the reader:

*L185. Based on these calculations and the measured rBC size distribution (discussed below), the SP2 mass and number concentrations measured at the rural site have been adjusted to correct for 25 and 30 % losses respectively since adjusting by size is not possible because the size of rBC cores including coating was not measured.*

**8. Line 216: Why does the greater concentration measured at the curbside suggest that these measurements are indicative of the city? Please explain in the main text.**

It's actually the similarity in concentrations, shown in Figure S4, that we think point to the validity of the curbside site as an indicator for the whole city. The text now reads

L214. *The comparison between the urban street-level and urban background eBC levels are very similar in time-series but with a lower daily maximum for the roof-top measurements (Fig. S4). This suggests that the curbside measurements are indicative for the city.*

**9. Line 266: Show statistical results to justify "statistically significant".**

We added the p-value from a two tailed Z-test on mass size distribution GMD.

*L265. The difference in mass GMD is statistically significant (P<0.01) and can be explained by different BC particle sources, air masses and coagulation.*

**10. Lines 275-293: Do the authors have any explanation about why 50 and 75 nm particles are unimodal, but 100 and 150 nm particles are bimodal?**

The APM measurements were taken during daytime, so there should exist both fresh and aged soot particles. The reason we didn't see bimodal distributions for the smaller sizes may have to do with the ambient total and soot size distributions (and chemical composition) but it is also harder to resolve effective densities of small particles which which requires smaller steps in voltage. Further, for fresh soot agglomerates, the smaller they are, the higher density they will have. This means that despite no coating the calculated effective density of fresh soot and Aitken mode aged particle may overlap.

**11. Line 307: I don't find the results of HYSPLIT trajectories in the main text and SI.**

These results are in section 3.6, Figure 7, and tables S2, S3 and S4.

**12. Line 340: How do the authors draw the statement that "approximately half of the mass at the urban site is due to long-range transport"? First, how "half of the mass" is estimated? Second, what is the evidence for "long-range transport" in Figure 8? Please explain this sentence in more detail in the main text.**

In Figure 8 we want to show the similarities in temporal pattern of pm1 non refractory components as well as the dissimilarity of eBC. The fact that approximately half of the mass is from long distance transport is deduced from the assumption of similar air masses (as is seen in non-refractory components) and the average concentrations at both sites. To help the reader, we added the mean and median concentrations to the figure. See below.

[Figure]

**13. Figure 8: Why does panel d have a different y-axis label from panel c?**

We only use the fractions of chemical species from the urban dataset. This is described in the methods, mentioned in the caption and also discussed with a previous reviewer, see below.

For the SP-AMS, there was confusion around the measured mass loadings (based on field calibration with 300 nm mobility diameter ammonium nitrate particles). The rural loadings were higher throughout the campaign, despite very similar time trends for all non-refractory species (see Figure 7). We found the concentration in the rural data well supported by DMPS (same site) and FIDAS (nearby background site). We do not have measurements for a similar evaluation of the urban SPAMS. Hence, the combined evidence suggests the urban absolute concentrations from SPAMS are too low, which we attribute to calibration issues. Notably this would not have been detected if we were not comparing with another SPAMS downwind (at the rural site) which was properly supported by auxiliary measurements.

This issue also affects Figure 9, where we also removed the absolute values now.

**14.Line 353: The authors showed the strong correlation between non-refractory PM1 concentration and thickly coated BC. What is the correlation between non-refractory PM1 concentration and non-coated BC (i.e., fresh BC)? If the latter correlation is also strong, the statement "the BC coatings are similar in composition to nonrefractory-PM1" is over-interpreted.**

From the delay time method of the SP2, described in section 2.2.2, the coating is divided into an either thickly or thinly coated fraction. Therefore, the thinly coated particles (fresh BC) is simply 1-thickly coated. Hence, there is an anticorrelation for this dataset.

**15.Line 368: I don't think a factor of 2.2-4 between urban and rural environments is "surprisingly low". The authors should consider using another word.**

We deleted this sentence.

**16.Line 388: As I mentioned in the major comment #1, the authors should rephrase this sentence.**

See answer to major comment 1 above.

**REVIEWER #2**

**General**

**The paper reports BC measurements in southern Sweden including source analyses and interpretations. In my opinion it suits well to the journal. This is the corrected version of the manuscript where the authors have replied to other reviewer's comments. For me the replies look fine, I didn't find much to correct any more. However, I did still find some things that I think should be corrected. They are in the detailed comments below.**

**Detailed comments and questions**

**L 83. " An Aerosol Particle Mass Analyzer (APM) was at a later stage deployed ..." Show the deployment period also in Fig. 1, just like for all the other instruments.**

The APM measurements were undertaken the year after. Including that in the figure would make it harder to resolve the intensive campaign dates, which we belive should be the main focus. We changed the caption to:

*Figure 1. Overview of the data and instruments used. Not shown in the figure are DMA-APM measurements of effective density during spring 2019.*

**L123. "Default corrections for filter scattering (Cref = 1.39) and loading effects were used (Weingartner et al., 2003)." The Weingartner et al. (2003) correction was developed for the older AE version, AE31. It is good for it. However, the AE33 calculates the loading corrections already during the measurements by using the dual-spot method and the loading correction function presented by Drinovec et al. So, how have you really used the Weingartner et al. correction? Correction on top of another correction? Or have you used the raw data and reprocessed that according to Weingartner et al.? If so, why, what is the justification? That makes no sense. According to the "Data availability" section you have uploaded the EBC data to EBAS. They do not accept such double corrections for level 0, level 1 or level 2. So, if you really have done this you should recorrect and resubmit the data to EBAS. Follow the ACTRIS recommendations and refer to them in the paper. After all, this is a "measurement report" so you should properly refer to the corrections you have done.**

Good point. The Cref value was displayed on recommendation by a previous reviewer. We did not do any post-correction. We have however cited the wrong paper, as pointed out. We removed this sentence to not confuse readers.

**L200. Caption of Figure 2. The word "trend" is wrong word here. Diurnal cycle is the right term. Trend is either increasing or decreasing in a longterm data. The same applies to the caption of Fig. S4, correct both.**

Thank you, corrected both.

**L214. "The AAE is similar between the sites, with small differences likely owing to different BC sources." Actually, the AAE values are different in a logical way. The AAE difference may be due to the BC size distribution and coating. At the rural site GMD is larger than at the urban site which may well lead to a lower AAE, in line with the simulations by Virkkula, Atmos. Meas. Tech., 14, 3707–3719, 2021.**

We agree and modified this discussion accordingly:

*L221. The AAE is slightly higher at the urban site, which could be due to the differences in BC size distribution and coating (Virkkula, 2021).*

**L215-219. Table 2. Add also the corresponding values of rBC from the SP-AMS. Discuss the differences between the rBC(SP2) and rBC(SP-AMS) also in the text, now there is no text about the rBC(SP-AMS). There are not many papers that would show both rBC(SP2) and rBC(SP-AMS), if any, I don't know (for you to find out...), so such a comparison would be very valuable.**

In our experience rBC as measured by SP-AMS using the default calibrant (at the time of this study it was Cabot inc "Regal Black", size selected at 300 nm with a DMA) is biased low. This has been shown (willis et al 2014) to be due to incomplete overlap between particle beam and laser beam used to vaporize the particles (only vaporised components are detected in SP-AMS), referred to as low particle "collection efficiency" (CE)

The problem is that the calibrant particles have higher CE than ambient soot particles, and thus the instrument sensitivity obtained from calibration is not valid for ambient soot. The particle beam width, and thus CE, is dependent on particle shape which varies considerably for soot depending on e.g. coatings (aged soot particles are round or round-ish, fresh soot is highly aspherical). The SP-AMS normally measures "particle ensemble" properties, as opposed to single particle properties. In the urban air there is a mixture of old (spherical) and fresh (aspherical) soot particles which complicates rBC quantification by SP-AMS, as single particle mixing state comes into play. For that reason we opted to focus on rBC as measured by SP2 (intensive campaigns), and eBC as measured by AE33 (environmental monitoring) in this work.

We do agree that the issue is not properly tackled by the SP-AMS community yet, but it is outside the scope of this manuscript. It would make an interesting companion paper though (single particle data om coating thickness IS available from the SP2, and morphology state can be inferred to some extent from the size resolved SP-AMS data as well).

**For the rBC modes in the table, give also their widths as GSD. And again, for both rBC(SP2) and rBC(SP-AMS).**

See answer above on rBC (SP-AMS). We added GSD to the table:

| | Rural | Urban |
|---|---|---|
| $M_{rBC}$ (1h), µg m$^{-3}$ | 0.06 ±0.05 | 0.15 ±0.11 |
| $M_{rBC}$ (1h), µg m$^{-3}$ | 0.22 ±0.16 | 0.49 ±0.45 |
| $N_{rBC}$ (1h), cm$^{-3}$ | 31 ±21 | 100 ±80 |
| $GMD_M$ (24h), nm | 168 ±12 | 141 ±11 |
| $GSD_M$ (24h), nm | 1.74±0.08 | 1.77±0.04 |
| $AAE_{370-950\ nm}$ (1h) | 1.13 ±0.12 | 1.24 ±0.26 |

**L275-276. "... mean mass of 0.062 ± 0.001 fg and 0.26 ± 0.01 fg, respectively, corresponding to effective densities of 0.95 ± 0.01 g cm-3 and 1.19 ± 0.02 g cm-3." How was this calculated? Give at least a reference, rather also the equation.**

This is described in section 2.2.6 with references.

**REVIEWER #3**

**The submitted manuscript focuses on the measurements of particulate matter in two locations, a rural and an urban location, in Sweden, during 2018. The authors presented measurements of rBC, eBC and investigated the chemical composition of non-refractory PM1. Furthermore, the authors conducted trajectory analysis to determine the influence of transport air pollution to the rural site.**

**The presented measurement report is within the scope of the ACP journal, and can be consider for a publication. The authors have addressed the comments of the previous reviewers, but there are still some comments than need to be addressed.**

**Comments**

**• In Ln 320 and Figure S1, the authors presented source apportionment analysis, but they did not discuss in the methods section the methods/tools used. In Figure S1, what do the Other Sources represent?**

Yes, it is described in the methods section, see below. Other sources are e.g. waste incineration, non-road-transport and fugitive emissions.

L200. The long-distance transported BC source contributions during the rural and urban measurement campaigns were estimated using the Lagrangian 1D-column chemistry transport model ADCHEM (Roldin et al., 2011; Roldin et al., 2019). ADCHEM was setup and run forward in time along pre-computed 14-days backward HYSPLIT air mass trajectories arriving 100 m a.g.l. at Hyltemossa, one new trajectory every hour. Source specific anthropogenic BC emissions along the trajectories were taken from the CAMS-GLOB-ANT v4.2 global emission inventory, which has a spatial resolution of 0.1°x0.1° (Granier et al., 2019). BC emissions from wildfires were estimated using the GFED4 emission inventory (Randerson et al., 2018).

**• In Ln 156, the authors used a dryer while sampling with the particle sizers. Did they account for any particle losses in their analysis?**

The particle losses are described in section 2.2.7. The dryer before the SP2 was accounted for by adding a 1.25 m equivalent pipe length (Wiedensohler et al., 2012). Either way, the losses in this section is not likely to have affected neither N or M concentrations in the range of the SP2.

Wiedensohler, A., et al.: Mobility particle size spectrometers: harmonization of technical standards and data structure to facilitate high quality long-term observations of atmospheric particle number size distributions, Atmospheric Measurement Techniques, 5, 657-685, 2012.

**• In Ln 241, a reference is missing regarding the measurement artifacts of the aethalometer due to the coating of BC core with non-absorbing material. Furthermore, one more reason for higher measurements is the coating of the BC core with absorbing organic aerosol, brown carbon. High relative humidity can also affect optical measurements.**

We added a reference comparing the AE33 specifically for this:

Kalbermatter, D. M., G. Močnik, L. Drinovec, B. Visser, J. Röhrbein, M. Oscity, E. Weingartner, A. P. Hyvärinen and K. Vasilatou: Comparing black-carbon- and aerosol-absorption-measuring instruments – a new system using lab-generated soot coated with controlled amounts of secondary organic matter, Atmos. Meas. Tech., 15, 561-572, 2022.

**• In Ln 214, the AAE is not similar between the two sites. The smaller core particles in the urban have higher AAE. Previous studies have discussed how the sizes of the core and the coating can affect the AAE.**

We have rephrased this, see answer to reviewer #2

**• In Ln. 205, the authors compare the measurements of the number concentration of the rBC although the SP2 number concentration measurements are not accurate, due to the lower size cut off. The authors should consider estimate the number concentration of the rBC by applying lognormal fit in the number size distributions.**

We compare the two sites using the same instrument and same cut-off size which we think is fair. The size distribution of BC mass equivalent cores is generally very broad in ambient air, consisting of several different emission sources. Since the diameter where the SP2 detection efficiency decreases is often close to where the number size distribution of BC is at its maximum, the actual total number concentration is very uncertain. We have added the geometric standard deviation to table 1 so the size distribution can be computed by interested readers.

**• The authors presented a comparison of the aethalometer and SP2 measurements. The authors should include in the manuscript the measurement uncertainty of the two methods.**

We have added the estimated relative uncertainty for the SP2

L99. *The overall relative uncertainty in mass concentration by the SP2 was estimated by Sharma et al. (2017) to be within 25-38 %.*

The estimation of the AE33 is not so easy to make. We have added a reference looking at AE51 instruments and the following text.

L132. *The relative uncertainty of the absorption coefficient measured by the aethalometer increases with lower BC mass concentrations, and has been shown to converge around 30 % for an older version of the instrument (Backman et al., 2017). The uncertainty of eBC also includes the uncertainty in the MAC value which was not estimated in the present study.*

**• In Ln.154 Please add references on the DMPS and the Hauke type DMAs.**

We have added the following reference:

Wiedensohler, et al: Mobility particle size spectrometers: harmonization of technical standards and data structure to facilitate high quality long-term observations of atmospheric particle number size distributions, Atmos. Meas. Tech., 5, 657-685, 2012.

**Minor Comments**

**• Figure 7: define CPH in the caption**

Fixed.

---

## Author Response (AR3)

We thank all reviewers and the editor for their comments. Our answers, in normal text, are presented below the comments, which are bold. Changes to the manuscript are in italic.

**REFEREE #3**

**The authors of the manuscript (acp-2022-156) have well addressed most of my comments in the last round of revision. However, there remain two comments that I highly recommend the authors consider addressing before publication. Once the comments are addressed, in my opinion, the manuscript is suitable to be accepted by the journal.**

**1. I still think the current introduction section is too general; instead, it should include more explanation about the variabilities of soot from different sources (the first paragraph). This is very important to the readers to understand why the authors focus on comparing absorption, size, and mixing state of BC between the sites.**

As suggested we have significantly reworked the introduction by expanding the sections on soot formation/transformation (which explains the variability from different sources), health and measurement techniques as well as added a few references.

The full section now reads as follows:

[revised manuscript text omitted]

**2. Add a time-series figure of BC and C4H9+ concentrations: The authors agreed that this could be a nice analysis, but decided not to add it to the current manuscript. Please add such a figure or explain why not adding. The figures can help readers understand the temporal variability of BC and HOA during a traffic plume. Adding the figure to the SI is fine.**

We have added the figure suggested by the referee and caption below to help the readers understand how the plume analysis was performed. This is now referenced in the methods section.

[Figure]

*Figure S2. An example of typical plumes of rBC number concentration (measured by SP2) and signal of the HOA proxy mass fragment C4H9+ (measured by SP-AMS). Units are arbitrary. Plume mass spectra were isolated by subtracting the mass spectra of the background before and after the plumes.*

**REFEREE #5**

**The manuscript reports the particulate matter measurements in two sites, a rural and an urban, located in Sweden. The authors discuss the importance of the long range PM transportation. A large suite of instrumentation was used for the measurements of eBC, rBC, the chemical characterization of non-refractory PM and their size distributions. The authors also presented trajectory analysis results.**

**I believe that the authors have adequately addressed the comments made by the reviewers**

**Minor comment:**

**1) in several figures the y-axis needs to be corrected. More specifically in Figures 2, 5, 9**

We have modified the figures as follows.

The error bars in Figure 2 was previously cut to better show the diurnal cycle. Since the std is symmetrical around the average all values can still be deduced from the original figure. However we have changed this now. New figure below.

[Figure]

Figure 2.

In Figure 5 we clarified that the y-axis is the number concentration, and arbitrary units.

[Figure]

Figure 5.

In figure 9 we adjusted the y-axis of the insets to match and clarified that the AMS concentrations are normalized:

[Figure]

Figure 9.

---

## Author Response (AR4)

We have adjusted all of the technical corrections (see below) in the uploaded file.

*Dear Erik Ahlberg and Coauthors,*

*I am happy to accept your manuscript for publication in ACP after the following nine technical corrections are incorporated into the revised manuscript.*

*Line 49: Replace "fuel used" with "used fuel"*

*Line 60: Delete "to" after "carcinogenic"*

*Line 62: Add a space before "Drawing"*

*Line 64: Replace "black carbon" with "BC"*

*Line 73: Add a space after "particles"*

*Lines 110-111: Replace "Single Particle Soot Photometer (SP2)" with "SP2"*

*Line 364: Delete one dot after "S9"*

*Line 26 (Supplement): Replace "table" with "Table"*

*Line 33 (Supplement): Replace "S7" with "S9"*